# Aging-regulated anti-apoptotic long non-coding RNA *Sarrah* augments recovery from acute myocardial infarction

D. Julia Trembinski[1,2], Diewertje I. Bink[3], Kosta Theodorou[1,2], Janina Sommer[1], Ariane Fischer[1],
Anke van Bergen [3], Chao-Chung Kuo [4], Ivan G. Costa [4], Christoph Schürmann[5],
Matthias S. Leisegang [2,5], Ralf P. Brandes [2,5], Tijna Alekseeva[6], Boris Brill[6], Astrid Wietelmann[7],
Christopher N. Johnson[8], Alexander Spring-Connell[9], Manuel Kaulich [10], Stanislas Werfel[2,11,16],
Stefan Engelhardt [2,11], Marc N. Hirt[2,12], Kaja Yorgan[2,12], Thomas Eschenhagen [2,12], Luisa Kirchhof[1],
Patrick Hofmann[1,2], Nicolas Jaé[1], Ilka Wittig[2,13], Nazha Hamdani[3,14], Corinne Bischof[1], Jaya Krishnan[1],
Riekelt H. Houtkooper[15], Stefanie Dimmeler[1,2] & Reinier A. Boon [1,2,3✉]

Long non-coding RNAs (lncRNAs) contribute to cardiac (patho)physiology. Aging is the major risk factor for cardiovascular disease with cardiomyocyte apoptosis as one underlying cause. Here, we report the identification of the aging-regulated lncRNA *Sarrah* (ENSMUST00000140003) that is anti-apoptotic in cardiomyocytes. Importantly, loss of *SARRAH* (*OXCT1-AS1*) in human engineered heart tissue results in impaired contractile force development. *SARRAH* directly binds to the promoters of genes downregulated after *SARRAH* silencing via RNA-DNA triple helix formation and cardiomyocytes lacking the triple helix forming domain of *Sarrah* show an increase in apoptosis. One of the direct *SARRAH* targets is NRF2, and restoration of NRF2 levels after *SARRAH* silencing partially rescues the reduction in cell viability. Overexpression of *Sarrah* in mice shows better recovery of cardiac contractile function after AMI compared to control mice. In summary, we identified the anti-apoptotic evolutionary conserved lncRNA *Sarrah*, which is downregulated by aging, as a regulator of cardiomyocyte survival.

[1] Institute for Cardiovascular Regeneration, Centre for Molecular Medicine, Goethe University Frankfurt am Main, Frankfurt am Main, Germany. [2] German Center for Cardiovascular Research (DZHK), Berlin, Germany. [3] Department of Physiology, VU University Medical Center, Amsterdam, the Netherlands. [4] Institute for Computational Genomics, Joint Research Center for Computational Biomedicine, RWTH Aachen University, Aachen, Germany. [5] Institute for Cardiovascular Physiology, Medical Faculty, Goethe University Frankfurt am Main, Frankfurt am Main, Germany. [6] Georg Speyer Haus, Institute for Tumor Biology and Experimental Therapy, Frankfurt am Main, Germany. [7] Max-Planck-Institute for Heart and Lung Research, Bad Nauheim, Germany. [8] Division of Clinical Pharmacology, Vanderbilt University Medical Center, Nashville, USA. [9] Department of Chemistry, Georgia State University, Atlanta, USA. [10] Institute of Biochemistry II, Goethe University, Frankfurt am Main, Germany. [11] Institute of Pharmacology and Toxicology, Technical University Munich, Munich, Germany. [12] Department of Experimental Pharmacology and Toxicology, University Medical Center Hamburg-Eppendorf, Hamburg, Germany. [13] Functional Proteomics, Medical School, Goethe University Frankfurt am Main, Frankfurt am Main, Germany. [14] Department of Cardiovascular Physiology, Ruhr University Bochum, Bochum, Germany. [15] Laboratory Genetic Metabolic Diseases, Academic Medical Center, Amsterdam, the Netherlands. [16] Present address: Department of Nephrology, Technical University of Munich, School of Medicine, Klinikum rechts der Isar, Munich, Germany. ✉email: Boon@med.uni-frankfurt.de

The majority of the genome is transcribed into RNA, but only about 2–3% of the transcriptome codes for proteins[1]. The remaining part of the transcriptome that does not code for proteins is called non-coding RNA. Many classes of non-coding RNAs have been described, including ribosomal RNAs, transfer RNAs, microRNAs and long non-coding RNAs (lncRNAs)[2]. The latter class is a heterogeneous group of non-coding RNAs that are by definition longer than 200 nucleotides. More than 100,000 lncRNAs have been described in humans[3] and several lncRNAs have been identified to play pivotal roles in homeostasis and disease[4]. The molecular pathways lncRNAs act on are diverse. For example, the lncRNAs Chast, Chaer, Fendrr and Mhrt all control cardiac function[5], but through very different mechanisms. For instance, Chast hinders cardiomyocyte autophagy by downregulating Pleckstrin homology domain-containing protein family M member 1 (Plekhm1), the overlapping gene on the opposite strand, thereby also driving hypertrophy[6]. Chaer promotes epigenetic reprogramming that induces cardiac hypertrophy by interfering with polycomb repressor complex 2 (PRC2)[7]. Although also involved in epigenetic control, Fendrr regulates heart development by modifying PRC2 occupancy at target genes[8,9]. Mhrt, another cardioprotective lncRNA, sequesters Brg1 from chromatin targets to prevent hypertrophic gene expression[10].

Aging is the predominant risk factor for cardiovascular disease[11]. In the heart, aging is characterized by an increase in stiffness, fibrosis and cardiomyocyte apoptosis, which are associated with an increase in heart failure. Several factors have been suggested to be of potential therapeutic use to counteract aging-induced cardiac dysfunction, by preventing cardiomyocyte apoptosis and inducing neovascularization. For instance, inhibition of miRNA-34a, which directly regulates PNUTS, reduces cardiomyocyte cell death and fibrosis during aging and after AMI[12]. However, the role of long non-coding RNAs in cardiac aging has not been thoroughly studied before.

Here, we describe the identification of the lncRNA Sarrah (short for SCOT1-antisense RNA regulated during aging in the heart), which is repressed during aging, and show that silencing Sarrah induces apoptosis and delays cardiac contractile force development in human engineered heart tissue (EHT). Mechanistically, Sarrah forms a DNA-DNA-RNA triplex with promoters of cardiac survival genes to recruit CRIP2 and activate gene expression. One of these target genes that confers its anti-apoptotic function is NRF2. Finally, we show that Sarrah can be used to therapeutically augment cardiac function after acute myocardial infarction in mice.

## Results

### Sarrah is an anti-apoptotic lncRNA downregulated by aging.
To assess which lncRNAs are regulated by aging in cardiomyocytes, we enzymatically dispersed cardiac cells in Langendorff-perfused hearts from young (8 weeks) and aged (18 months) mice. After differential centrifugation to separate cardiomyocytes from non-cardiomyocytes and RNA isolation, polyadenylated RNAs were sequenced by next generation sequencing on the Illumina HiSeq platform (Supplementary Fig. 1A). We identified 29,150 transcripts, of which 5439 were annotated as lncRNAs that are expressed in the cardiomyocyte fraction (Supplementary Fig. 1B). Of these lncRNAs, we selected 76 lncRNAs for which we found reliable reads when assessing expression in a genome viewer. We confirmed expression of these lncRNAs by qRT-PCR in the HL-1 mouse cardiomyocyte cell line and adult mouse cardiac tissue (Supplementary Fig. 1C).

One of the hallmarks of cardiac aging is loss of cardiomyocytes by apoptosis. To assess whether any of the identified lncRNAs

regulates apoptosis, we employed an siRNA-based screening approach to reduce expression levels of all 76 lncRNAs identified above in combination with a caspase-3/7 activity-based apoptosis assay (Fig. 1a). This assay showed that the lncRNA with the largest effect on apoptosis in HL-1 cardiomyocytes was a transcript annotated as ENSMUST00000140003[13]. As this was the most potent effect we observed, we further focused on this lncRNA and named it Sarrah (SCOT1-antisense RNA regulated during aging in the heart) since it is transcribed from the antisense locus of the OXCT1 gene encoding the enzyme SCOT1. To establish whether regulation of apoptosis by Sarrah could be an evolutionary conserved mechanism, we searched for homologous transcripts in humans, pigs and rats using publicly available sequencing and annotation databases (http://genome.ucsc.edu/[14–18]) and found transcripts in the ontogenic loci with small stretches of conserved sequences (Fig. 1b, Supplementary Fig. 1D, supplementary Table 1A). We verified that Sarrah is a non-coding transcript using the CPAT algorithm[19] for the human and mouse sequences (Supplementary Table 1B–C). Sarrah is present in several cardiac cell types, including cardiomyocytes (Fig. 1c, Supplementary Fig. 3A) and repressed during aging of the heart (Supplementary Fig. 3B), as confirmed by qRT-PCR in a separate cohort of mice (Fig. 1d).

The main clinical presentation of heart disease in the elderly is heart failure with preserved ejection fraction (HFpEF). We therefore measured Sarrah in hearts of a rat HFpEF model[20] and found a significant reduction of Sarrah levels in rats that display a HFpEF phenotype compared to those without HFpEF phenotype (Supplementary Fig. 3C).

We aimed to confirm the initial findings of the siRNA-based approach with a second loss-of-function strategy. Therefore, we used LNA-DNA-based antisense oligonucleotides that induce RNase H-mediated cleavage of the targeted RNA in the nucleus. These so-called GapmeRs were transfected in vitro and Sarrah levels were measured by qRT-PCR, showing a significant decrease in Sarrah levels in comparison to transfection with control GapmeRs, both in mouse and human cardiomyocytes (Supplementary Fig. 4A). Consistently, inhibition of Sarrah significantly induces caspase activity in mouse and human cardiomyocytes (Fig. 1e). Next, we confirmed the induction of apoptosis by Sarrah silencing with a flow cytometry-based annexin V/7-AAD assay in mouse cells (Supplementary Figs. 4B and 12). Conversely, lentiviral SARRAH overexpression in human cardiomyocytes (Supplementary Fig. 4C) resulted in a profound decline in caspase activity (Fig. 1f). These results confirm the anti-apoptotic role of Sarrah in cardiomyocytes.

Together, the results demonstrate that Sarrah is an aging-regulated, conserved lncRNA that is potentially anti-apoptotic in cardiomyocytes.

### Sarrah enhances cardiomyocyte survival and contractility.
Next, we assessed the role of Sarrah for cardiomyocyte contractility in detail in in vitro models. Contractility measurements with primary neonatal rat cardiomyocytes showed a reduced contraction amplitude after Sarrah knockdown with GapmeRs (Fig. 2a, Supplementary Fig. 2A). Also the maximal contraction and relaxation velocities were reduced (Fig. 2a), which reflect diminished cardiac contractile function after conditions such as AMI and in HFpEF, respectively. It must, however, be noted that we cannot exclude that these effects on contractility and relaxation are secondary to effects on viability.

To translate the findings about the role of Sarrah for cardiomyocyte apoptosis and contractility from rodent cells to human tissue, we used human EHT organoids (Fig. 2b). TUNEL staining of the EHTs revealed an increase in apoptosis in the

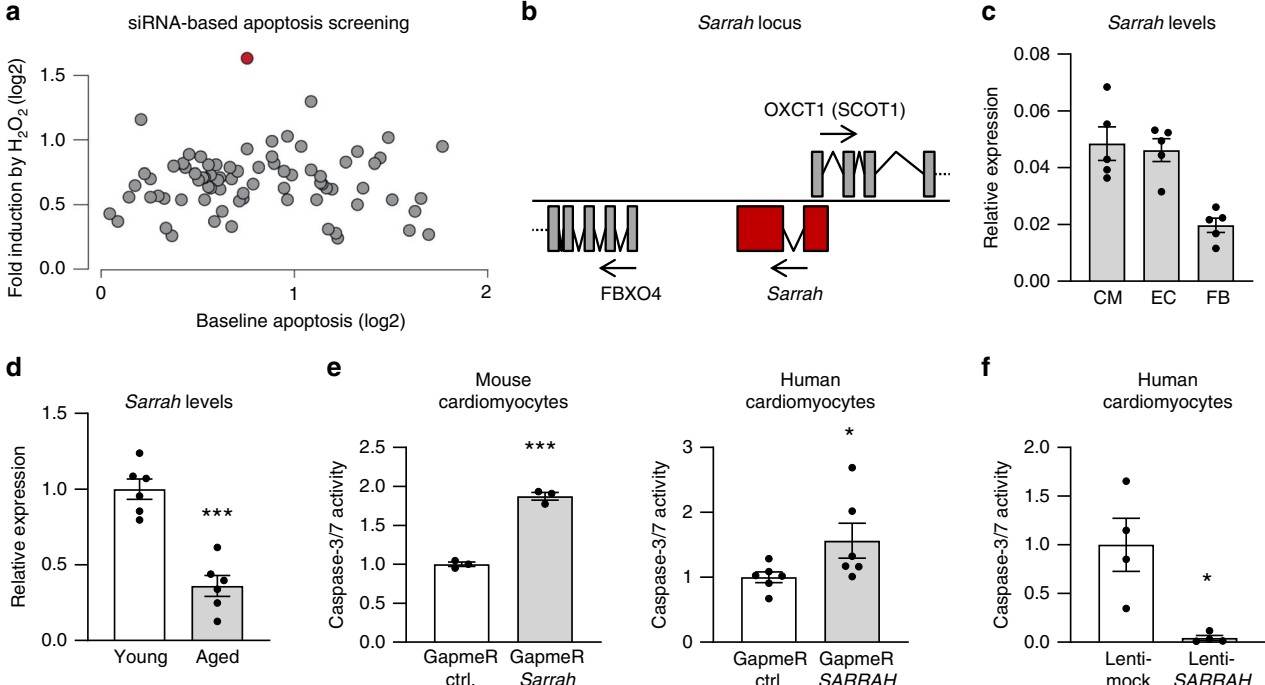

**Fig. 1 Sarrah is an evolutionary conserved, anti-apoptotic lncRNA in cardiomyocytes downregulated during aging. a** Two different siRNAs against each of the 76 cardiomyocyte-enriched lncRNAs from Supplementary Fig. 1c were transfected in HL-1 cells ($n = 3$), the average for both is displayed. The lncRNA highlighted in red corresponds to lncRNA Sarrah. Apoptosis levels were determined in standard cell culture conditions or after induction with 100 µM $H_2O_2$ by measuring caspase-3/7 activity. **b** The genomic Sarrah locus overlaps with the OXCT1 gene encoding SCOT1. Its transcription start site lies within the first OXCT1 intron. **c** Three different cell types (cardiomyocytes (CM), endothelial cells (EC) and fibroblasts (FB)) were isolated from the hearts of 12-week-old mice. RNA was isolated and Sarrah levels were determined by qRT-PCR ($n = 5$; SEM). **d** Sarrah downregulation during aging was confirmed by qRT-PCR with RNA from total young and aged mouse heart tissue ($n = 6$; SEM; ***$t$-test $p = 0.00005$). **e** Caspase-3/7 activity was measured in GapmeR-transfected mouse (HL-1 cell line) and human (primary cells) cardiomyocytes to confirm the increase in apoptosis upon Sarrah knockdown (HL-1: $n = 3$; SEM; ***$p = 0.0001$); hCM: $n = 6$; SEM; *$t$-test $p = 0.0486$). **f** Caspase-3/7 activity was measured in SARRAH overexpressing primary human cardiomyocytes ($n = 4$; SEM; *$t$-test $p = 0.0286$).

SARRAH-silenced organoids (Fig. 2c), further confirming an anti-apoptotic role of SARRAH. Moreover, the SARRAH-silenced organoids developed contractile force with a severe delay and to a lesser extent than control organoids (day 12 vs. day 5, Fig. 2d). Fractional shortening was also reduced (Fig. 2d). In this model system, defects in contraction are likely caused by loss of cardiomyocytes, rather than effects on contractile properties of individual cardiomyocytes. Together, these results show that SARRAH regulates human cardiomyocyte survival and contractility.

**Sarrah activates gene expression via triplex formation.** Next, we investigated the mechanism via which Sarrah acts. Cis gene regulation has been reported for several lncRNAs[21], but Sarrah does not seem to regulate apoptosis by regulating gene expression in cis (Supplementary Fig. 5). Neither FBXO4 nor c5orf51, two nearby genes on the same strand, were significantly regulated after Sarrah silencing (Supplementary Fig. 5A). OXCT1, the gene partially overlapping with the Sarrah locus on the antisense strand, showed a slight downregulation after Sarrah knockdown in mouse and human cardiomyocytes (Supplementary Fig. 5B). However, SCOT1 (the enzyme encoded by OXCT1) enzymatic activity remained almost unaffected by Sarrah silencing (Supplementary Fig. 5C). Furthermore, the induction of apoptosis after Sarrah silencing is not likely to be caused by a reduction of OXCT1, as silencing OXCT1 did not result in apoptosis (Supplementary Fig. 5D). Sarrah silencing did also not affect Fbxo4 or c5orf51 mRNA levels in mouse and human cardiomyocytes (Supplementary Fig. 5B). We therefore conclude that the anti-

apoptotic effects of Sarrah are not conferred via cis gene regulation.

To determine genes regulated by SARRAH, we performed a microarray analysis with human cardiomyocytes treated with a GapmeR control and a GapmeR targeting SARRAH, respectively (Fig. 3, Supplementary Fig. 6). In total, 501 genes ($p < 0.05$) were downregulated while 961 genes ($p < 0.05$) were upregulated after SARRAH silencing (Supplementary Fig. 6A). Gene set enrichment analysis revealed pathways affected by SARRAH silencing (Fig. 3a), mostly apoptosis-related pathways, confirming that the dominant effect of SARRAH silencing is an increase in apoptosis. We confirmed the regulation of some of the most significantly regulated genes by qRT-PCR (Supplementary Figs. 6B–C).

Since Sarrah is predominantly localized in the nucleus (Fig. 3b) in mouse and human cardiomyocytes and associates with chromatin as indicated by an RNA-immunoprecipitation against histone H3 (Fig. 3c), we hypothesized that it might directly regulate gene expression. The Triplex Domain Finder software[22] identified a high incidence of triple helix prone motifs in promoters of genes regulated after SARRAH silencing. We therefore hypothesized that SARRAH might form RNA-DNA triple helix structures. To test this, we used the Triplex Domain Finder software to analyze the human SARRAH sequence with regard to RNA stretches that are capable of binding genomic DNA in promoters of regulated genes via triple helix formation (Fig. 3d).

One SARRAH region was significantly predicted to form triplexes (Fig. 3e) via Hoogsteen base pairing in promoters of 134

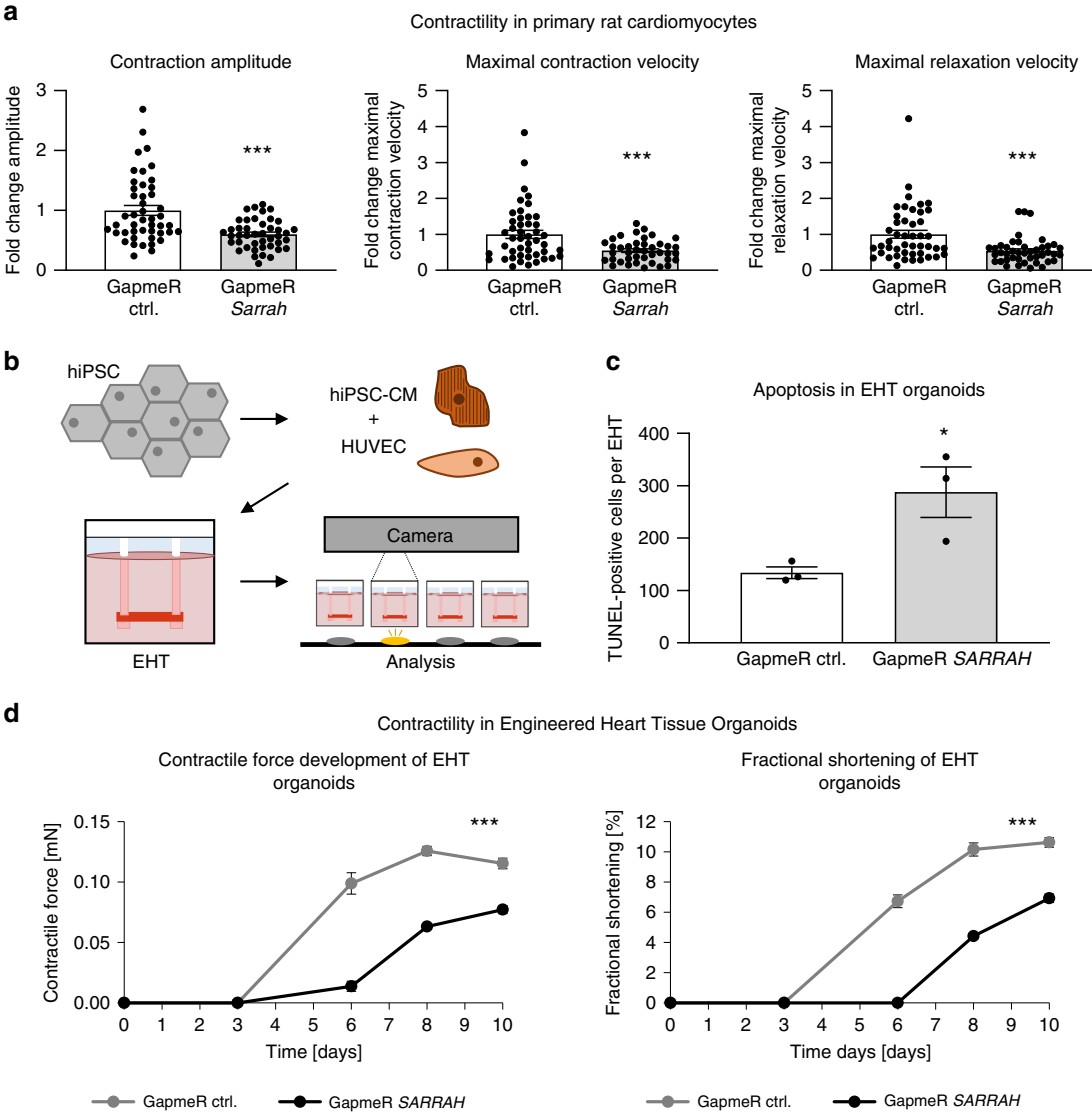

**Fig. 2 Sarrah is required for contractility of human and rat cardiomyocytes. a** Primary cardiomyocytes were isolated from neonatal rats. Contractility was measured using the IonOptix Myocyte Contractility Recording System and analyzed using the IonWizard software ($n = 43$ and $46$; SEM; ***t-test $p = 0.00003$ for contraction amplitude, $p = 0.00045$ for maximal contraction velocity, $p = 0.00041$ for maximal relaxation velocity). **b** Scheme depicting the generation of EHTs and the experimental setup (hiPSC human induced pluripotent stem cell, CM cardiomyocytes, HUVEC human umbilical vein endothelial cells, EHT engineered heart tissue organoid). **c** Apoptosis of EHTs was measured by terminal deoxynucleotidyl transferase dUTP nick end labeling (TUNEL)-positive nuclei per EHT ($n = 4$; SEM; *t-test $p = 0.036$). **d** EHTs consisting of hiPSC-derived cardiomyocytes and HUVECs were treated with 4 μM GapmeRs for 2 days. Contractile force and fractional shortening were assessed on days 0, 3, 6, 8, and 10 using the EHT contractility analysis instrument and the corresponding software ($n = 8$; SEM; Force: ***two-way ANOVA treatment $p < 0.0001$, $F = 273.63$; ***two-way ANOVA time $p < 0.0001$, $F = 306.72$; Fractional shortening: ***two-way ANOVA treatment $p < 0.0001$, $F = 205.65$; ***two-way ANOVA time $p < 0.0001$, $F = 219.76$).

genes that were downregulated after *SARRAH* silencing ($p = 0.0033$; Supplementary Fig. 6A, highlighted in red) as shown for the exemplary *GPC6* promoter (Fig. 3f). In contrast, no *SARRAH* binding site was identified in the promoters of upregulated genes ($p = 0.42$). These results suggest that *SARRAH* may bind gene promoters to activate transcription. We validated the regulation of some of the 134 genes from the triple helix prediction that are potentially involved in apoptosis by qRT-PCR (Supplementary Fig. 6C). Interestingly, gene ontology analysis of all 134 genes revealed that the disease categories associated with these genes were "cardiovascular" and "aging" (Supplementary Table 2).

For experimental verification of the formation of triple helices between *SARRAH* and gene promoters, we used nuclear magnetic resonance (NMR) spectroscopy to characterize base pair formation between the proposed human *SARRAH* triple helix domain

as a single-stranded RNA oligonucleotide and its predicted binding site in the human promoter of *GPC6* as a double-stranded DNA oligonucleotide.

For Watson–Crick base pairings, imino proton NMR peaks are typically observed between 12 and 14 ppm (Fig. 3g). Under slightly acidic conditions, cytosine can be protonated at the N3 position[23], which facilitates cytosine-rich RNA binding to DNA duplexes[24]. As anticipated, at neutral pH the addition of equal molar ratio of cytosine-rich RNA to the DNA duplex did not produce additional peaks in the imino proton region. However, reducing the pH to slightly acidic conditions resulted in the appearance of several new peaks (Fig. 3g, red and orange). The spectra are consistent with the formation of Hoogsteen base pairing and support the formation of a $C^{+}$-G-C triplex structure[25]. We confirmed that the new peaks were the result of

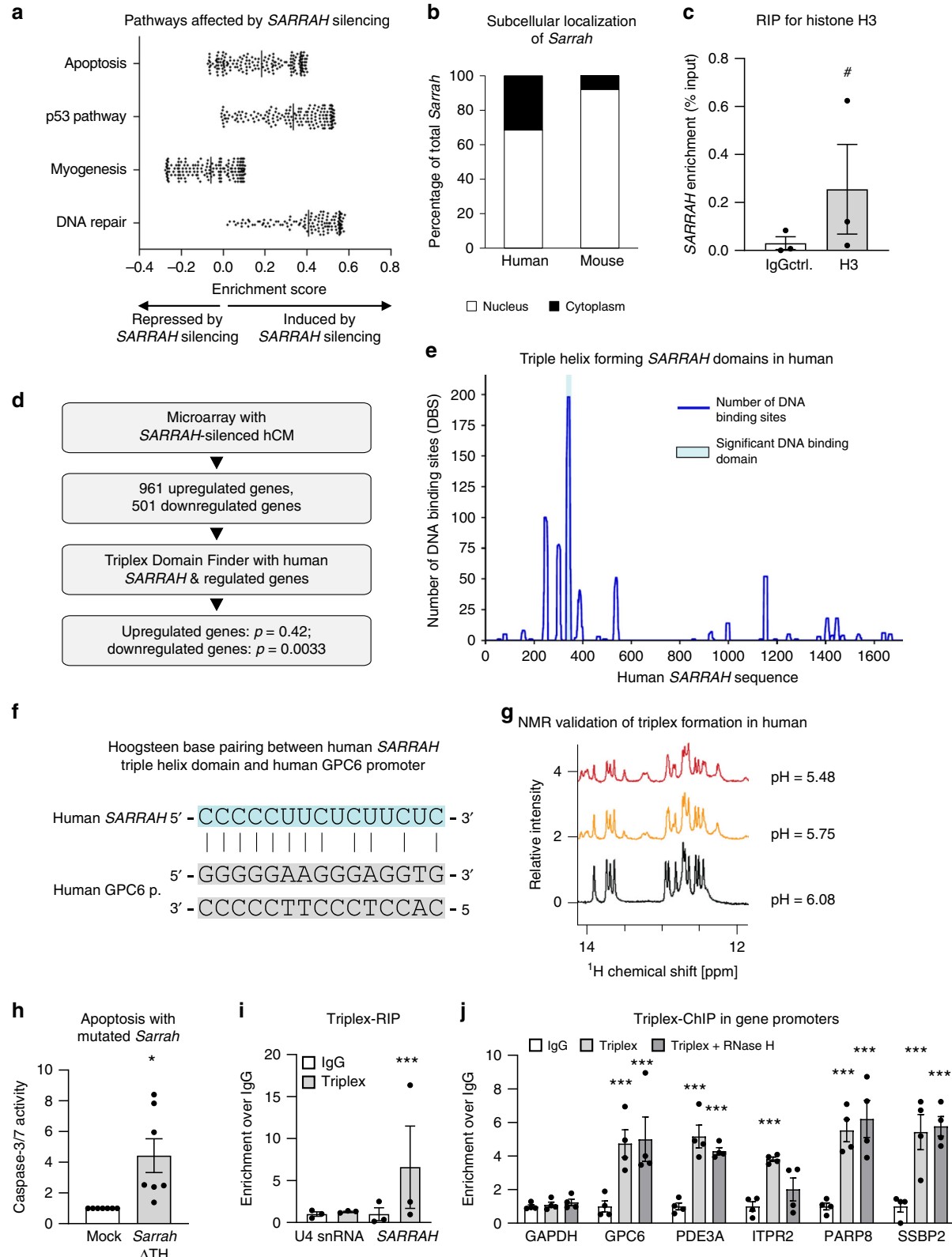

triplex formation and not a subpopulation of unique DNA conformations, an RNA hairpin or RNA-RNA duplex by repeating the experiment with isolated DNA and isolated RNA (Supplementary Fig. 7A). These findings indicate that the *SARRAH* triple helix domain is able to interact with DNA duplexes in the predicted promoter regions to form a parallel DNA-DNA-RNA triplex.

Additionally, we confirmed that triple helix formation is a conserved mechanism, as mouse *Sarrah* is also predicted to form triple helices with promoters of genes that were downregulated after *Sarrah* silencing ($p = 0.0012$; Supplementary Fig. 7B, Supplementary Table 3). Mouse *Sarrah* was predicted to target 162 gene promoters. Forty percent of the predicted *Sarrah* target genes overlap between human and mouse, among others the

**Fig. 3 SARRAH activates gene expression via triple helix formation with gene promoters. a** Microarray data of primary human cardiomyocytes treated with GapmeR control or GapmeR *SARRAH* were analyzed for differentially regulated pathways using gene set enrichment analysis. **b** Human (primary) and mouse (HL-1 cell line) cardiomyocytes were fractionated into their cytoplasmic and nuclear portions and *Sarrah* levels were measured by qRT-PCR in both fractions ($n = 3$). **c** RNA-immunoprecipitation with an anti-total histone H3 antibody was performed in primary human cardiomyocytes. *SARRAH* levels were measured by qRT-PCR ($n = 3$; SEM; # $t$-test $p = 0.0504$; IgG *immunoglobulin G*). **d** Flow chart illustrating the procedure of microarray analysis of primary *SARRAH*-silenced human cardiomyocytes (hCM), identification of *SARRAH* DNA binding domains and DNA binding sites. **e** The human *SARRAH* sequence was assessed with regard to pyrimidine-rich regions capable of DNA binding via triple helix formation using the Triplex Domain Finder software. **f** Scheme indicating Hoogsteen base pairing between the human *SARRAH* triple helix domain and the human *GPC6* promoter. **g** [1]H spectra of the *SARRAH* binding site in the human *GPC6* promoter as a DNA duplex 15mer in the presence of the human *SARRAH* triple helix domain as equal molar single-stranded RNA, as analyzed by nuclear magnetic resonance (NMR). **h** Using the CRISPR/Cas9-mediated approach outlined in Supplementary Fig. 7e the *Sarrah* triple helix domain was excised from the endogenous gene locus in mouse cardiomyocytes (HL-1 cell line). Apoptosis was quantified as caspase-3/7 activity ($n = 7$; SEM; *$t$-test $p = 0.0156$). **i** RNA-immunoprecipitation with the S9.6 anti-DNA-RNA-hybrid antibody was performed in crosslinked primary human cardiomyocytes. Levels of U4 snRNA as a negative control and *SARRAH* were measured by qRT-PCR (**n** = 3; SEM; ***$t$-test $p = 0.0003$; IgG *immunoglobulin G*). **j** Chromatin-immunoprecipitation with the S9.6 anti-DNA-RNA-hybrid antibody was performed in crosslinked primary human cardiomyocytes. Sonicated DNA fragments were used for qRT-PCR to quantify triplex formation in gene promoters ($n = 4$; SEM; ***two-way ANOVA IgG vs. RNA-DNA hybrid: $p = 0.00017$ for GPC6, $p = 0.0044$ for ITPR2 and $p < 0.0001$ for all other promoters; two-way ANOVA IgG vs. RNA-DNA hybrid + RNase H: $p = 0.00089$ for PDE3A and $p < 0.0001$ for all other promoters; $F = 11.24$ for variable "promoter" and $F = 52.8$ for variable "antibody"; IgG *immunoglobulin G*).

mouse *GPC6* promoter (Supplementary Fig. 7C). We experimentally confirmed that mouse *Sarrah* forms a triple helix with the predicted region in the mouse *GPC6* promoter by NMR (Supplementary Fig. 7D). The spectra indicate that the mouse *Sarrah* triple helix region binds to the *GPC6* promoter via Hoogsteen bonds.

To functionally confirm the importance of the *Sarrah* triple helix domain, we used a CRISPR/Cas9-mediated approach to delete the endogenous region in mouse cardiomyocytes (Fig. 3h, Supplementary Figs. 7E–G). We confirmed the deletion on RNA level by qRT-PCR (Supplementary Fig. 7G). Importantly, *OXCT1* mRNA levels were not significantly affected by the deletion of the *Sarrah* triple helix domain (Supplementary Fig. 7G). Caspase activity was significantly higher in cells lacking the *Sarrah* triple helix domain compared with control cells (Fig. 3h), consistent with the increase in caspase activity after *Sarrah* silencing. Notably, mRNA levels of several selected *Sarrah* target genes (*GPC6, PDE3A, ITPR2, PARP8, SSBP2*) were downregulated (Supplementary Fig. 7G), similar to the effects of *SARRAH* silencing. Finally, we performed immunoprecipitation experiments with an antibody that recognizes RNA-DNA hybrids[26] to first confirm that *SARRAH* physically interacts with genomic DNA (Fig. 3i). Unlike negative control U4 snRNA, *SARRAH* was significantly enriched over IgG. To identify triple helix-associated gDNA structures, we performed chromatin immunoprecipitation (ChIP) using the same antibody combined with RNase H digestion, which degrades RNA-DNA-structures such as R loops without affecting triple helices[27]. All five measured *SARRAH* target gene promoters (*GPC6, PDE3A, ITPR2, PARP8, SSBP2*) were enriched over IgG control as opposed to *GAPDH* promoter (Fig. 3j). Except for *ITPR2*, *SARRAH* target gene promoters were also significantly enriched over IgG control after RNase H digestion. These findings validate that the majority of predicted *SARRAH* target genes forms triple helices in their promoters.

To assess how *SARRAH* regulates apoptosis in a more unbiased manner, we performed a proteome profiler assay with antibodies against 35 apoptosis-related proteins with human cardiomyocyte cell lysates after knockdown of *SARRAH* or control cell lysates (Fig. 4a). Interestingly, several anti-apoptotic proteins were reduced, among them catalase, bcl-2, heme oxygenase 1, and bcl-x. The expression of all of these proteins is controlled by the anti-oxidant NRF2/Keap1 transcription factor pathway[28]. Strikingly, NRF2 (*NFE2L2*) is a predicted direct *SARRAH* target and NRF2 expression is reduced on both mRNA and protein level after knockdown of *SARRAH* (Fig. 4b). Moreover, reactive oxygen species (ROS) levels are increased in cardiomyocytes

following knockdown of *SARRAH*, suggesting reduced activity of the anti-oxidant NRF2/Keap1 system (Fig. 4c). Finally, we assessed whether restoration of NRF2 signaling by lentiviral overexpression of NRF2 rescues the pro-apoptotic effect of *SARRAH* depletion (Fig. 4d). Indeed, NRF2 overexpression partially negates the induction of apoptosis as a result of *SARRAH* depletion, indicating that NRF2 is one of the main mediators of the cardioprotective effects of *SARRAH*.

Together, these results indicate that *Sarrah* functions via triple helix formation with gene promoters, thereby inducing transcription of cardiac survival genes, including NRF2, and inhibiting apoptosis of cardiomyocytes.

**SARRAH recruits CRIP2 and p300.** Since lncRNAs have been extensively described to interact with proteins to participate in chromatin modifications[29–31], we performed RNA affinity purification of endogenous *Sarrah* from mouse cardiomyocyte lysate to identify protein interaction partners of *Sarrah* that potentially facilitate transcriptional activation of *Sarrah* target genes (Fig. 4e–g). Mass spectrometry analysis revealed the cardiac transcription factor cysteine-rich protein 2 (CRIP2)[32,33] as most significantly enriched interaction partner of *Sarrah* (Fig. 4g), which we validated in an RNA immunoprecipitation (RIP) experiment in human cardiomyocytes (Fig. 4h). Since CRIP2 has been reported to interact with the transcriptional co-activator p300 in human cardiomyocytes[34], we investigated whether *SARRAH* binds to p300, and identified a significant interaction between *SARRAH* and p300 (Fig. 4h). P300 is an acetyltransferase that acetylates histone H3 on lysine 27 (H3K27ac) to activate transcription[35]. Therefore, we performed RIP experiments with an antibody against H3K27ac, which confirmed that *SARRAH* associates with open chromatin (Fig. 4h). This corroborates our hypothesis that *SARRAH* activates gene expression via triple helix formation with gene promoters.

These results reinforce our findings that *Sarrah* facilitates gene transcription by forming triple helices with gene promoters and thereby recruiting the transcription factor CRIP2 and the transcriptional co-activator p300.

**Sarrah overexpression improves cardiac function.** To assess whether *Sarrah* overexpression could also functionally inhibit apoptosis in a setting where cardiomyocyte apoptosis is detrimental, we first used adeno-associated virus particles of serotype 9 (AAV9) to overexpress *Sarrah* or GFP as control in 18-month old mice. In hearts of these aged mice, with ongoing age-induced

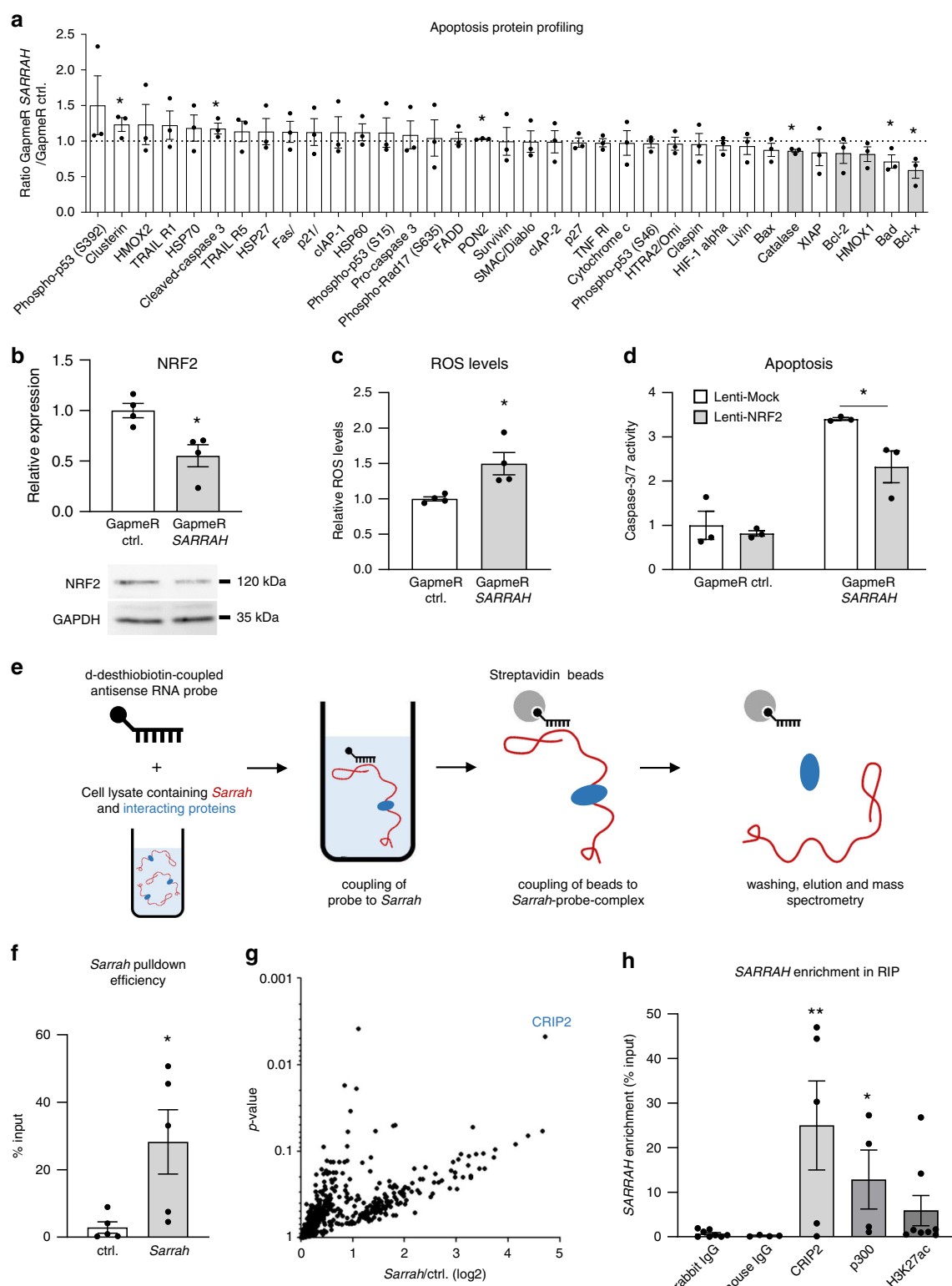

apoptosis, *Sarrah* overexpression significantly diminished apoptosis, indicating that the reduction of *Sarrah* levels in aged mice functionally contributes to cardiomyocyte cell death in vivo (Fig. 5a). As a next model where cardiomyocyte apoptosis plays a detrimental role, we employed an acute myocardial infarction (AMI) model. First, we determined the expression levels of *Sarrah* by qRT-PCR (Fig. 5b) in the infarcted region and the border region in hearts of mice subjected to AMI at 1, 3, 7, and 14 days after AMI as well as in sham-operated control mice. These

experiments showed that *Sarrah* is strongly downregulated in the infarcted and border regions after AMI, a response that is present in cardiomyocytes, endothelial cells and cardiac fibroblasts (Supplementary Fig. 8A). One of the hallmarks of the infarcted myocardium is hypoxia, which raised the hypothesis that *Sarrah* may be downregulated due to hypoxia in the myocardium. To test this, we examined whether hypoxia regulates *Sarrah* expression in cardiomyocytes in vitro. Both mouse and human cardiomyocytes were treated with the hypoxia-mimicking compound

**Fig. 4 SARRAH regulates apoptosis by induction of NRF2 and recruits CRIP2 and p300 to activate gene transcription. a** A Proteome Profiler assay (R&D Systems) was performed with GapmeR-transfected human cardiomyocytes to assess levels and phosphorylation status of apoptosis-related proteins upon *SARRAH* knockdown. Black bars depict downregulated NRF2 target genes ($n = 3$; SEM; *t-test $p = 0.001$ for Catalase, $p = 0.0046$ for PON2, $p = 0.016$ for Bad, $p = 0.017$ for Bcl-x, $p = 0.024$ for Clusterin, $p = 0.026$ for cleaved caspase-3). **b** NRF2 mRNA levels were measured by qRT-PCR in human cardiomyocytes after transfection with GapmeRs to silence *SARRAH* or control GapmeRs (top panel; $n = 4$; SEM; *t-test $p = 0.014$). NRF2 protein levels were measured by Western blotting. GAPDH served as loading control ($n = 3$). **c** Reactive oxygen species were measured using a CM-H2DCFDA probe in human cardiomyocytes after transfection with GapmeRs to silence *SARRAH* or control GapmeRs ($n = 4$; SEM; *t-test $p = 0.021$). **d** Caspase-3/7 activity was measured in human cardiomyocytes that lentivirally overexpress NRF2 or mock-transduced cells that were transfected with either control GapmeRs or GapmeRs targeting *SARRAH* ($n = 3$; SEM; *two-way ANOVA $F = 6.8$, $p = 0.0266$). **e** Scheme depicting the RNA pulldown approach used to identify proteins interacting with endogenous *Sarrah*. 200 pmol of biotinylated scrambled oligo or two biotinylated *Sarrah* antisense oligos were added to HL-1 cell lysate, coupled to streptavidin beads and eluted. **f** *Sarrah* pulldown efficiency was determined by qRT-PCR of eluted samples (displayed as % input; $n = 5$; SEM; *t-test $p = 0.0317$). **g** Volcano plot showing all proteins identified by mass spectometry analysis that are enriched in *Sarrah* pulldown as compared to pulldown with a scrambled oligo. CRIP2, the hit with the highest enrichment, is highlighted. **h** RNA-immunoprecipitations with antibodies against CRIP2 (rabbit antibody), p300 (mouse antibody) and histone acetylation H3K27ac (rabbit antibody) were performed in primary human cardiomyocytes. *SARRAH* levels were measured by qRT-PCR ($n = 4$, 5 and 8; SEM; *t-test $p = 0.029$ for p300; one-way ANOVA for CRIP2 $p = 0.024$; IgG *immunoglobulin G*).

deferoxamine mesylate (DFO) or subjected to hypoxia (1% and 0.2% $O_2$, respectively; Supplementary Figs. 8B–C). These results showed that *Sarrah* is robustly downregulated by hypoxic conditions.

To test the hypothesis that *Sarrah* repression after AMI contributes to cardiomyocyte apoptosis and hampered recovery of contractile function, we employed AAV9-mediated cardiac overexpression of *Sarrah* in combination with AMI followed by echocardiography and MRI analysis of cardiac function (Fig. 5c). AAV9-mediated *Sarrah* overexpression significantly increased *Sarrah* levels in the myocardium (Fig. 5d). The wall motion score index (WMSI), a parameter that reflects contractile function, showed that cardiac contractile recovery was significantly enhanced in the *Sarrah* overexpressing group compared to controls (Fig. 5e). Moreover, *Sarrah* expression levels at day 14 after AMI significantly correlated with changes in WMSI (Supplementary Fig. 9A). Consistently, left ventricular wall thickness (Fig. 5f) and stroke volume (Fig. 5g) were significantly increased upon *Sarrah* overexpression. Measurements of ejection fraction with magnetic resonance imaging as well as echocardiographic measurements comparably demonstrated that the recovery of contractile function at 14 days after AMI was significantly enhanced in the *Sarrah* overexpressing group compared to the control group (Fig. 5h, Supplementary Figs. 6B–E).

To more closely assess the effects of *Sarrah* overexpression on cardiac phenotype, we performed histological stainings on heart sections. The number of apoptotic nuclei, which was visualized by TUNEL- and DAPI-co-staining, was significantly reduced in *Sarrah* overexpressing animals in the infarct region while no difference was observed in the border region (Fig. 5i). Interestingly, when we stained phospho-histone H3 (PH3)-positive nuclei, we found a significant increase in proliferation of non-cardiomyocytes while cardiomyocyte proliferation was unaffected (Fig. 5j). Since fibrosis was decreased by *Sarrah* overexpression, we hypothesized that the proliferating cells were endothelial cells rather than fibroblasts. To verify this hypothesis, we co-stained the endothelial cell marker isolectin B4 with phospho-histone H3 and detected significantly more proliferating endothelial cells in *Sarrah* overexpressing hearts (Fig. 5k). Fibrosis staining on serial sections revealed that *Sarrah* overexpressing hearts had a significantly smaller infarct size than controls (Fig. 5l).

Since AAV9 particles do not directly target endothelial cells, we speculated that *Sarrah* in cardiomyocytes may indirectly affect endothelial cell proliferation. Therefore, we co-cultured *SARRAH* overexpressing human cardiomyocytes with human umbilical vein endothelial cells (HUVECs; Supplementary Fig. 10A) and analyzed HUVEC proliferation by flow cytometry. In accordance

with our in vivo findings, HUVECs proliferated more when co-cultured with *SARRAH* overexpressing human cardiomyocytes in vitro (Supplementary Figs. 10B and 12). Conversely, when *SARRAH* was depleted from human cardiomyocytes by GapmeR-treatment, HUVECs proliferated less (Supplementary Fig. 10C). Moreover, using conditioned medium of *SARRAH*-silenced human cardiomyocytes on HUVECs increased apoptosis (Supplementary Fig. 10D) while in vitro sprouting capacity was reduced (Supplementary Fig. 10E) in comparison to culturing in conditioned medium of control-transfected cardiomyocytes.

Together, these results demonstrate that *Sarrah* overexpression reduces cardiomyocyte apoptosis, induces endothelial cell proliferation and augments cardiac contractile function after AMI in mice, thereby substantially contributing to cardiac AMI recovery.

## Discussion

Our data demonstrate that the lncRNA *Sarrah* is suppressed in aged and infarcted hearts. Silencing of *Sarrah* causes apoptosis in cardiomyocytes and augmentation of *Sarrah* in vivo improves cardiomyocyte survival and cardiac contractile function after AMI. Mechanistically, *Sarrah* interacts with DNA via triple helix formation in promoter regions of genes that are activated by *Sarrah* to recruit transcriptional activators that switch on a gene expression program that induces cell survival (supplementary figure 11). Increasing survival of myocardium and simultaneously inducing angiogenesis is regarded as a promising therapeutic strategy for AMI, for example by blocking caspase activity[36] and stimulating VEGF signaling[37], respectively. Overexpression of *Sarrah* is capable of both. However, these conclusions from the in vivo experiments are based on exogenous overexpression of *Sarrah* and therefore do not allow to firmly conclude that *Sarrah* is a necessary factor to protect the heart from functional deterioration in the context of aging or myocardial infarction, a pathological outcome that cannot be comprehensively assessed in in vitro models alone. Importantly, *Sarrah* is evolutionary conserved and silencing of human *SARRAH* induces apoptosis of cardiomyocytes and reduces proliferation of co-cultured endothelial cells, suggesting that the potential therapeutic effects of *Sarrah* overexpression for AMI treatment may be translated from the mouse model to human AMI patients.

In analogy to transcription factors or microRNAs that can control expression of an entire set of target genes that act in synergy to elicit a certain cellular response, we propose that the set of genes that are induced by *Sarrah* provides the anti-apoptotic and pro-survival effects. One of the key target genes identified here is the anti-oxidant transcription factor NRF2, which is known to be anti-apoptotic and cardioprotective[38]. Indeed, overexpression of NRF2 partially rescues the induction of

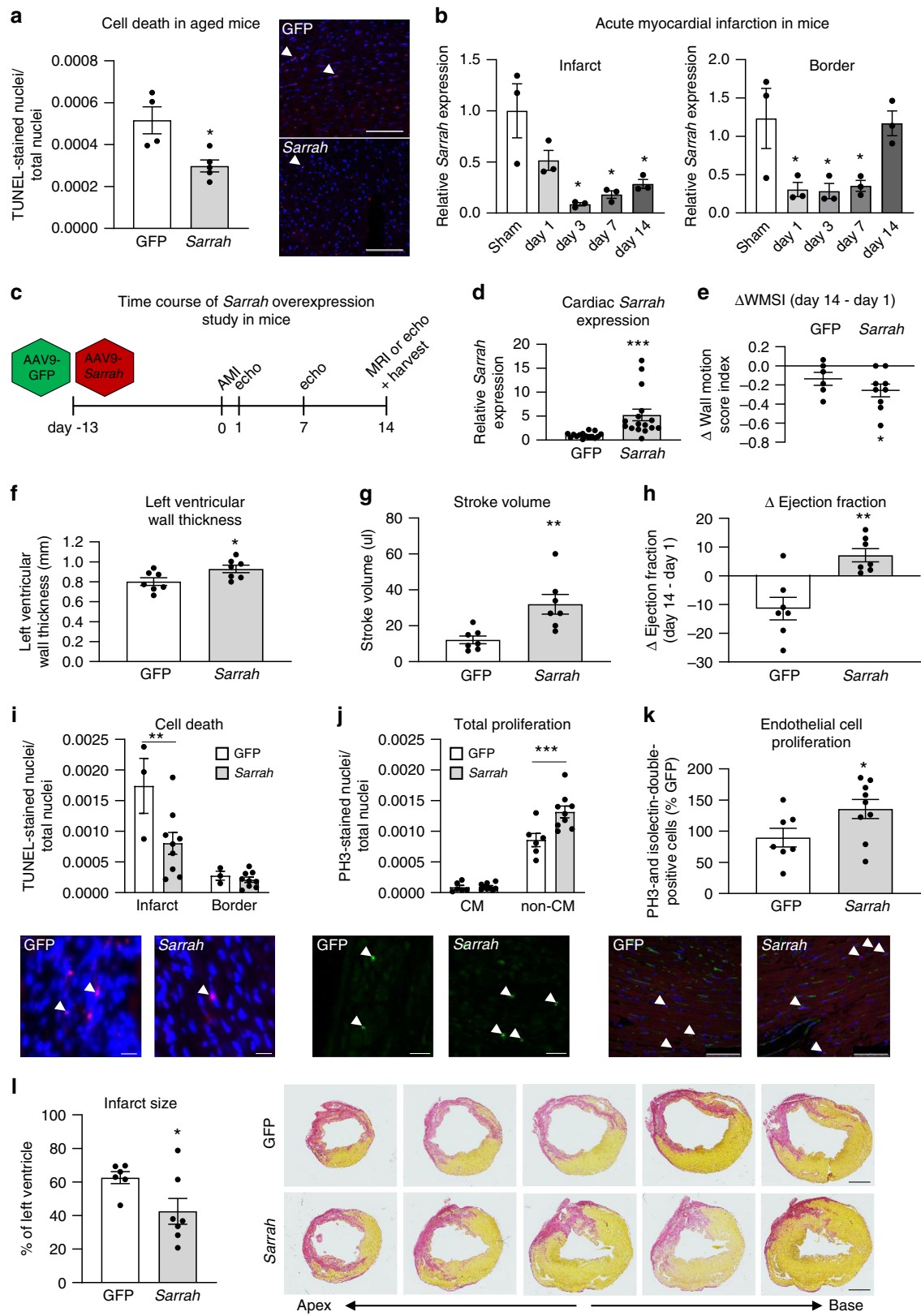

apoptosis after silencing *SARRAH*. However, other *SARRAH* targets likely contribute to the phenotype controlled by *SARRAH*. This gene expression program not only induces cardiomyocyte survival, but also induces endothelial cell proliferation in a paracrine manner.

Our in vivo and in vitro data suggest that cardiomyocyte-resident *Sarrah* regulates proliferation of endothelial cells in a paracrine manner. Non-coding RNAs, especially microRNAs, are known to be transferred between cells via microvesicles or protein complexes[39], but due to its nuclear localization and chromatin

**Fig. 5 *Sarrah* overexpression in mice improves recovery from acute myocardial infarction. a** Adeno-associated virus serotype 9 (AAV9)-green fluorescent protein (GFP) or AAV9-cytomegalovirus (CMV)-*Sarrah* virus was injected intravenously into 18-month-old mice three weeks prior to sacrifice. Apoptosis was measured by terminal deoxynucleotidyl transferase dUTP nick end labeling (TUNEL)-positive nuclei per total number of nuclei ($n = 4$ and 5; SEM; *$t$-test $p = 0.016$; representative images are shown; scale bars are 100 μm). **b** *Sarrah* levels in the infarct and border regions of mouse hearts after myocardial infarction (AMI) surgery were measured by qRT-PCR on days 1, 3, 7, and 14 and compared with sham-operated hearts ($n = 3$; SEM; *One-way ANOVA, Infarct: $F = 8.0$, $p = 0.0018$ for day 3, $p = 0.0038$ for day 7, $p = 0.0093$ for day 14, Border: $F = 5.8$, $p = 0.0275$ for day 1, $p = 0.0246$ for day 3, $p = 0.036$ for day 7). **c** AAV9-green fluorescent protein (GFP) or AAV9-cytomegalovirus (CMV)-*Sarrah* virus was injected intravenously into mice two weeks prior to AMI. Cardiac function was analyzed by echocardiography at 1, 7 and 14 days and by MRI at 14 days after AMI. **d** *Sarrah* levels of mouse hearts after AMI were measured by qRT-PCR ($n = 15$ and 16; SEM; *$t$-test $p < 0.0001$). **e** Wall motion score index (WMSI) was assessed from echocardiographic measurements. Delta WMSI values at day 14 refer to day 1 ($n = 6$ and 9; SEM; *$t$-test $p = 0.0156$). **f** Left ventricular wall thickness was assessed from magnetic resonance imaging (MRI) at day 14 after AMI ($n = 7$; SEM; *$t$-test $p = 0.0346$). **g** Stroke volume was calculated as the difference between end diastolic and end systolic volumes, which were assessed from MRI at day 14 after AMI ($n = 7$; SEM; **$t$-test $p = 0.0023$). **h** Cardiac contractile function was assessed by echocardiography and MRI. Displayed are delta ejection fraction values on day 14 (MRI) after AMI in comparison to day 1 (echocardiography; $n = 7$; SEM; **$t$-test $p = 0.0064$). **i** Apoptosis was measured by terminal deoxynucleotidyl transferase dUTP nick end labeling (TUNEL)-positive nuclei per total number of nuclei on paraffin sections of AMI hearts ($n = 3$ and 9; SEM; **two-way ANOVA, $p = 0.0072$, $F = 31.33$ for variable "region", $F = 5.38$ for variable "treatment"; representative images are shown; scale bars are 20 μm). **j** Total proliferation was measured by phosphorylated histone H3 (PH3)-positive nuclei per total number of nuclei. ($n = 6$ and 9; SEM; ***two-way ANOVA, $p = 0.0004$, $F = 225.5$ for variable "cell type", $F = 7.84$ for variable "treatment"; representative images are shown; scale-bars: 20 μm; CM cardiomyocytes). **k** Endothelial cell proliferation was measured by PH3- and isolection-double-positive nuclei ($n = 7$ and 9; SEM; *$t$-test $p = 0.042$; representative images are shown; scale bars are 75 μm). **l** Serial sections were stained using sirius red and infarct size was assessed by circumference of the infarcted region as percentage of the left ventricle ($n = 6$ and 7; SEM; *$t$-test $p = 0.047$; representative images are shown; scale-bars:1 mm). White arrows indicate positive cells.

association, *Sarrah* is not very likely to be directly transferred from cardiomyocytes to endothelial cells. We therefore hypothesize that *Sarrah* induces the expression of genes that stimulate, in a paracrine manner, the proliferation of endothelial cells, by promoting the secretion of either exosomes or soluble factors from cardiomyocytes, both of which have been shown to stimulate endothelial cell proliferation[40–42]. In vivo, proliferation of endothelial cells could also be secondary to increased cardiomyocyte survival and concomitant increase in perfusion demand, but our in vitro results point to a more direct cardiomyocyte-endothelial cell cross-talk, as nutrient and oxygen availability is not affected in that system.

Together, the data presented here highlight a pathway that may be exploited to treat cardiovascular disease, especially in the elderly.

## Methods

**Animal experiments**. Animal experiments were performed in accordance with the principles of laboratory animal care as well as German national laws and have been approved by the ethics committee of the regional council (Regierungspräsidium Darmstadt, Hesse, Germany). All animals were obtained from Charles River. Mice were kept in individually ventilated cages (Tecniplast) at 12:12 h-light/dark cycles at 21–24 °C and 45-60% humidity. Water and ssniff R/M-H complete feed (ssniff Spezialdiäten, Soest, Deutschland) were fed ad libitum. The mouse *Sarrah* sequence (ENSMUST00000140003) driven by a cytomegalovirus (CMV) promoter was cloned into a single-stranded AAV-vector backbone. Recombinant AAV9 vectors were produced using helper plasmid co-transfection and purified in iodixanol gradients as described elsewhere[43]. Vectors containing the green fluorescent protein (GFP) sequence served as controls. $6 \times 10^{11}$ AAV9 particles were injected into tail veins of male 12-week-old C57Bl/6 mice two weeks before AMI surgery was performed by LAD coronary artery ligation as described elsewhere[44]. Echocardiographic assessment of cardiac contractility based on EF and WMSI was performed 1, 7 and 14 days after AMI using a Vevo 770 imaging system (VisualSonics). For magnetic resonance imaging analysis see below. Mouse hearts were harvested on day 14 after AMI by a researcher blinded to the treatment.

**Magnetic resonance imaging**. Cardiac magnetic resonance imaging (MRI) measurements were performed on a 7.0 T Bruker Pharmascan 70/16, equipped with a B-GA 9 s HP (760 mT/m) gradient system, using a $^1$H planar receive-only surface coil (SUC 300 LNA) with an inner diameter of 20 mm together with a transmit/receive volume resonator in transmit-only mode (RES 89/72QSN) and the IntraGate™ self-gating tool (Bruker, Ettlingen, Germany). The mice were measured two weeks after AMI under volatile isoflurane (2.0%) anesthesia. The measurements were based on the gradient echo method (repetition time = 6.2 ms; echo time = 1.6 ms; field of view = 2.20 × 2.20 cm; slice thickness = 1.0 mm; matrix = 128 × 128; oversampling = 100). The imaging plane was localized using scout images showing the 2- and 4-chamber view of the heart, followed by acquisition in

short axis view, orthogonal on the septum in both scouts. Multiple contiguous short axis slices consisting of 9 or 10 slices were acquired for complete coverage of the left ventricle. Magnetic resonance imaging data were analyzed using freely available analysis software Segment v1.8 (http://segment.heiberg.se) to give left ventricle functional parameters[45].

**Cardiomyocyte isolation and RNA sequencing**. Male 8-week (young) and 18-month (aged) old mice were purchased from Charles River and hearts were isolated and subjected to Langendorff perfusion and digestion as described[12,46]. Cardiac myocytes were separated from other cells by density centrifugation and snap frozen. RNA was isolated as described further below and used for RNA sequencing as described before[47]. Gene expression was estimated using Cufflinks version 2.1 with default parameters.

**Isolation of different cell types from mouse hearts**. Hearts from male mice were minced and treated with dispase II (Merck), collagenase II (Thermo Fisher Scientific) and DNase I (Sigma-Aldrich). The digested hearts were sifted through a cell strainer (PluriSelect, 100 μm pore size), washed with DMEM with 10% FBS and with PBS and centrifuged at $30 \times g$ and 4 °C for 5 min to pellet cardiomyocytes. The supernatant was pooled, sifted through a cell strainer (PluriSelect, 40 μm pore size), washed with DMEM with 10% FBS and centrifuged at $300 \times g$ and 4 °C for 5 min. Endothelial cells were isolated from the pellet by incubation with a CD31-antibody (1.25 μl per heart, Thermo Fisher Scientific #14-0311-82) and Dynabeads (sheep anti-rat IgG, 6 μl per heart, Invitrogen #11035) for 30 min at 4 °C under rotation. The supernatant was incubated in a cell culture dish at 37 °C and 5% $CO_2$ for 40 min to let the fibroblasts adhere. All cell fractions were resuspended in QIAzol (QIAgen) for RNA extraction.

**Cell culture**. Mouse cardiomyocytes (HL-1 cells) were purchased from Sigma-Aldrich and cultured in Claycomb medium (Sigma-Aldrich) supplemented with 10% fetal bovine serum (FBS; Thermo Fisher Scientific), 100 μg/ml penicillin/streptomycin (Roche), 0.1 mM norepinephrine (Sigma-Aldrich) and 2 mM GlutaMAX (Thermo Fisher Scientific). Cell culture dishes were coated with 0.02% gelatin (Merck) and 0.1% fibronectin from human plasma (Sigma-Aldrich) for cultivation of HL-1 cells.

Sarcomeric alpha-actinin and slow muscle myosin-positive human cardiomyocytes (hCMs) were purchased from PromoCell and cultured in Myocyte Growth Medium (PromoCell) with the supplier's Supplement Mix C and 100 μg/ml penicillin/streptomycin.

Neonatal rat cardiomyocytes were isolated from P1 Sprague Dawley rat pups according to the Neonatal Heart Dissociation protocol by Miltenyi Biotec[48] using the gentleMACS Dissociator (Miltenyi Biotec). They were cultured in plating medium (DMEM high glucose (BioConcept) supplemented with 17% EBS M199 (BioConcept), 10% horse serum (HS; BioConcept), 5% FBS, 4 mM L-glutamine (Thermo Fisher Scientific) and 100 μg/ml penicillin/streptomycin) on glass coverslips coated with poly-L-lysine (Sigma-Aldrich) and 1 mg/ml type I bovine collagen (Sigma-Aldrich) for 24 h. The day after cell isolation, the medium was changed to maintenance medium (DMEM high glucose (BioConcept) supplemented with 10% EBS M199 (BioConcept), 1% HS (BioConcept), 1% HS and 4 mM L-glutamine (Thermo Fisher Scientific)).

Human umbilical vein endothelial cells (HUVECs) were purchased from Lonza and cultured in endothelial basal medium supplemented with EGM SingleQuots (Lonza) and 10% FBS (Invitrogen).

HEK293T cells were cultured in DMEM supplemented with 10% heat-inactivated FBS, D-glucose, pyruvate and penicillin/streptomycin.

All cells were cultured at 37 °C and 5% $CO_2$ and counted with a NucleoCounter NC-200 (ChemoMetec). Hypoxia was induced by cultivation at 1% (HL-1 cells) or 0.2% (hCMs) $O_2$ at 37 °C and 5% $CO_2$ for 24 h or treatment with 300 μM deferoxamine mesylate (DFO; Sigma-Aldrich) for 24 h.

Cells were transfected at 70% confluency with 100 nM (HL-1 cells) or 25 nM (hCMs) LNA GapmeRs (Exiqon) or 50 nM siRNAs (QIAgen) 24 h after seeding using Lipofectamine RNAiMAX (Thermo Fisher Scientific) according to the manufacturer's guidelines in a mixture of 15% serum reduced Opti-MEM medium (Thermo Fisher Scientific) and 85% Claycomb medium supplemented with 10% fetal bovine serum and 2 mM glutamine (HL-1 cells) or 85% Myocyte Growth Medium (hCMs). LNA GapmeR negative control A (Exiqon) and an siRNA against firefly luciferase (Sigma-Aldrich) were used as controls. Experiments were performed 48 h after transfection unless otherwise indicated. For LNA GapmeR treatment of neonatal rat cardiomyocytes, no transfection reagent was used. Instead, 325 nM LNA GapmeRs were added to maintenance medium and cardiomyocyte contractility was measured 72 h later. LNA GapmeR and siRNA sequences are listed in Supplementary Table 4.

For co-culturing hCMs with HUVECs, hCMs were seeded on transwell inserts with 1 μM pore size (Greiner Bio-One) and transfected in Myocyte Growth Medium. HUVECs were seeded in cell culture plates and cultured in endothelial basal medium (EBM; Lonza) supplemented with EGM SingleQuots (Lonza) and 10% FBS. Twenty-four hours after transfection, inserts with hCMs were washed with phosphate-buffered saline (PBS) and transferred to HUVECs so that both were cultured in EBM. Alternatively, conditioned medium from transfected hCMs was transferred to HUVECs 24 h after transfection. Experiments were performed 72 h after hCM transfection.

Lentiviral vectors (pKLV2.2, Addgene #72666; pLenti4/V5-DEST, Thermo Fisher Scientific #V49810) were transfected into human embryonic kidney (HEK) 293T cells using psPAX2 (Addgene #12260) as packaging plasmid and pMD2.G (Addgene #12259) or pCMVΔR8.91 as envelope expressing plasmid. Lentiviral transduction of HL-1 cells with pKLV2.2-constructs together with a pLentiCRISPRv2_neo construct or hCMs with pLenti4-constructs was performed for 24 h. Cells were used for downstream applications within six days after transduction. Lentivirus with an NRF2-construct as described by Fledderus et al.[49] was transduced in hCMs for 24 h. Cells were transfected with GapmeRs seven days after transduction.

**Quantitative real-time PCR**. Total RNA was isolated using the RNeasy Mini Kit and the RNase-Free DNase Set by QIAgen (cellular and tissue samples), Direct-zol RNA MicroPrep or MiniPrep Kit by Zymo Research (cellular samples) according to manufacturer's instructions. DNase digestion was performed during all RNA extractions. 100 to 1000 ng of total RNA were reversely transcribed into cDNA using random hexamer primers (Thermo Fisher Scientific) and MulV or Super-Script Reverse Transcriptase (Applied Biosystems). For quantitative real-time PCR (qRT-PCR), 6.25 to 25 ng cDNA per reaction were used with Fast SYBR Green Master Mix (Applied Biosystems) in StepOne Plus or Viia7 instruments (Applied Biosystems). Gene expression levels were normalized to glyceraldehyde-3-phosphate dehydrogenase (GAPDH), ribosomal protein, large, P0 (RPLP0) or hypoxanthine-guanine phosphoribosyltransferase 1 (HPRT1) levels and evaluated using to the $2^{-\Delta CT}$ method. Primer sequences are listed in supplementary table 5.

**Caspase assay**. Caspase-3/7 activity in mouse and human cardiomyocytes and HUVECs was assessed using the Apo-ONE Homogeneous Caspase-3/7 kit (Promega) according to the manufacturer's instructions 48 h after treatment. Shortly, caspase substrate Z-DEVD-R110 was diluted 1:100 in Apo-ONE Homogenous Caspase 3/7 buffer and incubated with the cells for 1 h at 37 °C. For induction of apoptosis, $H_2O_2$ was added to the cells at a final concentration of 100 μM 4 h before substrate addition. Fluorescence was measured at the GloMax-Multi Detection System (Promega) at 521 nm wavelength.

**Proteome profiler assay**. The Proteome Profiler™ Array Human Apoptosis Array Kit (R&D Systems) was used according to the manufacturer's instructions 48 h after GapmeR treatment. Shortly, arrays were blocked for 1 h with array buffer and then incubated with cell lysates overnight at 4 °C. Arrays were incubated with Detection Antibody Cocktail for 1 h and subsequently with Streptavidin-HRP for 30 min at RT. Chemi Reagent Mix was used for visualization with the Amersham Imager 600 (GE Health care). Band intensities were quantified in ImageQuant TL (GE Health care).

**ROS measurement**. ROS levels in hCMs were assessed using a CM-H2DCFDA probe (Invitrogen). CM-H2DCFDA was incubated for 1.5 h at 37 °C. Cells were washed with HBSS. For induction of ROS levels, tert-butyl $H_2O_2$ was added to the cells at a final concentration of 500 μM 1 h before the measurement. Fluorescence was measured at the FLUOstar galaxy (BMG) at 521 nm wavelength.

**Immunoblotting**. hCMs were washed with ice-cold PBS and lysed in TX-100 RIPA buffer with protease inhibitor (Halt™ Protease Inhibitor Cocktail (100×)), phosphatase inhibitor (Halt™ Phosphatase Inhibitor Cocktail (100×)) and benzonase nuclease (Santa Cruz Biotechnology). Protein concentrations were determined with Pierce BCA Protein Assay Kit (Thermo Scientific) and lysates were treated with 5× sample buffer (312.5 mM Tris, pH = 6.8, 50% glycerol, 0.37 mM bromphenol blue, 347 mM SDS, 2.5% β-mercaptoethanol). Equal protein amounts were separated by SDS-polyacrylamide gel electrophoresis and transferred to 0.2-μM nitrocellulose membranes (GE Healthcare). Membranes were blocked in 5% BSA for 1 h. Primary antibodies (NRF2 sc-722, Santa Cruz, 1:500; GAPDH 14C10, Cell Signaling, 1:10,000) were diluted in blocking solution and incubated overnight (4 °C). HRP-conjugated secondary anti-rabbit antibodies (P0448, Dako, 1:5000) were incubated for 1 h at RT. ECL detection (Merck Millipore) was used for visualization with the Amersham Imager 600 (GE Healthcare). Band intensities were quantified in ImageQuant TL (GE Healthcare). The uncropped blots are depicted in the source data file.

**Histology**. For histological analysis, mouse hearts were fixed in 4% PBS-buffered formaldehyde overnight and embedded in paraffin. TUNEL (cell death) and sirius red (fibrosis) stainings on mouse heart sections were performed as described elsewhere[44]. Infarct size was quantified as ratio of infarct length to left ventricle circumference on five serial sections as described elsewhere[50]. Phosphorylated histone H3 (PH3; cell proliferation) was visualized by immune fluorescence staining using an anti-PH3 (Ser10) antibody (1:100 in PBS, Merck #6570). TUNEL staining of EHT was performed on whole specimens. Positively stained cells are displayed as absolute numbers per total number of cell nuclei or per organoid, respectively.

**EHT organoids**. This investigation conforms to the principles outlined by the Declaration of Helsinki and the Medical Association of Hamburg. All materials were taken with informed consent of the donors. All procedures involving the generation and analysis of hiPSC lines were approved by the local ethics committee in Hamburg (Az PV4798, 28 October 2014). Human induced pluripotent stem cells (hiPSCs) were differentiated to cardiomyocytes as described elsewhere[51]. Shortly, hiPSCs were obtained by reprogramming of fibroblasts from a healthy female donor with Yamanaka factors using the CytoTune-iPS Sendai Reprogramming Kit (Life Technologies). Onset of spontaneous beating was observed between days 8 and 10. After 17 days, hiPSC-derived cardiomyocytes were dissociated into single cells and subjected to EHT generation.

Each EHT organoid was composed of a mixture of one million hiPSC-derived cardiomyocytes and 300,000 GFP-expressing HUVECs. The cells were mixed with matrigel basement membrane matrix, fibrinogen and thrombin at a final volume of 100 μl in rectangular casting molds in a 24-well format. After 90 minutes, EHT organoids were transferred to new cell culture dishes with DMEM supplemented with 10% inactivated horse serum, penicillin/streptomycin, insulin, aprotinin and 10 μM ROCK inhibitor Y-27632. From day 0 to day 2, 4-μM LNA GapmeRs were added to the medium. From day 2 on, EHT medium was changed three times per week and supplemented with Y-27632 until day 7. After 2 days, first organoid areas started to beat spontaneously, and after 6 days, all organoids beat coherently. EHT movements were recorded three times per week under sterile conditions. Contractile forces were analyzed with the EHT contractility analysis instrument (EHT Technologies GmbH) as described elsewhere[52]. After 22 days in culture, EHT organoids were fixed in a phosphate-buffered formaldehyde solution (pH = 7; 4%) stabilized with methanol for 24 h and stored at 4 °C.

**Rat cardiomyocyte contractility measurements**. After isolation (see section Cell culture), 250,000 rat cardiomyocytes were plated on coated glass coverslips. 24 h later, plating medium was changed to maintenance medium and LNA GapmeRs were added at a final concentration of 325 nM. Contractility of individual cardiomyocytes was measured using the IonOptix Myocyte Contractility Recording System following the manufacturer's instructions. Briefly, cardiomyocytes were placed in a chamber mounted on the stage of an inverted microscope and perfused with warm (37 °C) modified tyrode buffer (137 mM NaCl, 5 mM KCl, 15 mM glucose, 1.3 mM $MgSO_4$, 1.2 mM $NaH_2PO_4$, 20 mM HEPES, 1 mM $CaCl_2$, pH = 7.4) and field stimulated at a frequency of 1 Hz. Contractility was recorded and analyzed using the IonWizard software.

**Microarray and bioinformatic analysis**. Microarray analysis of human cardiomyocyte RNA from LNA GapmeR control- and LNA GapmeR SARRAH-treated samples was performed using GeneChIP Human Exon 1.0 ST arrays (Affymetrix). Gene set enrichment analysis was used as published[53] and gene ontology analysis was performed using the Database for Annotation, Visualization and Integrated Discovery (DAVID) bioinformatics resources (https://david-d.ncifcrf.gov/).

**Triplex domain finder analysis**. Triplex formation between Sarrah and gene promoters was predicted using the Triplex Domain Finder (TDF)[54] with the human and mouse Sarrah sequence (OXCT1-AS1 and ENSMUST00000140003, respectively) as well as gene sets up- and downregulated by SARRAH silencing in the microarray. Sarrah DNA binding domains (DBD) were identified by the

promoter test. Promoters are defined as regions 1 kb upstream of the transcription start site based on GENCODE version 24 and genome hg38 for human data or GENCODE version 4 and genome mm10 for mouse data. The TDF was parametrized to indicate all DBD with at least 120 DNA target sites and an enrichment with $p$-value < 0.05.

**Sequence homology alignment**. Sequence alignment of human and mouse *Sarrah* was performed using the LALIGN DNA:DNA tool of the FASTA Sequence Comparison software published by the University of Virginia[50] with default parameters and scoring matrix +5/−4.

**RNA immunoprecipitation with protein-binding antibodies**. For RNA immunoprecipitation, 5 µg antibody (serotype control antibody rabbit IgG: Millipore #12-370; serotype control antibody mouse IgG: Santa Cruz #2025; antibody against total H3: abcam #1791) were coupled to 50 µl protein G beads per condition (ThermoFisher #21349), that had been washed three times in binding buffer (50 mM Tris-HCl (pH = 8), 150 mM NaCl, 0.05% NP-40 and 1 mM EDTA) at 4 °C under rotation overnight. The next day, two confluent 15-cm-dishes of human cardiomyocytes per condition were washed with PBS, pelleted at 4 °C and 500 × *g* and lysed in 100 µl 50 mM Tris-HCl (pH = 8), 150 mM NaCl, 0.5% NP-40 and 1× protease inhibitor for 15 min on ice. Supernatants were cleared by spinning for 10 min at 4 °C and 10,000 × *g* and 1 ml lysis buffer without NP-40 per condition was added. For immunoprecipitation, beads were washed three times in binding buffer and 1 ml lysate was incubated with the antibody-bead complex for 4 h at 4 °C under rotation. Beads were washed three times with binding buffer again, treated with proteinase K (NEB #P8107S) for 30 min at 50 °C and resuspended in QIAzol for phenol/chloroform RNA extraction together with 10% input. DNase digestion was performed during RNA extraction.

**RNA immunoprecipitation with DNA:RNA-hybrid antibodies**. To identify DNA-associated RNAs, human cardiomyocytes were crosslinked with UV light at 254 nm. Nuclei were isolated with the truCHIP™ Chromatin Shearing Kit (Covaris, USA) according to the manufacturer's protocol, but without sonication. Dilution buffer (20 mM Tris-HCl, pH = 7.4; 100 mM NaCl; 2 mM EDTA; 0.5% Triton X-100; protease inhibitors) was added to lysates that were subsequently pre-cleared with 20 µl DiaMag protein A- and protein G-coated magnetic beads (Diagenode, Seraing, Belgium) for 30 min at 4 °C. The samples were incubated overnight at 4 °C with 5 µg anti-DNA-RNA hybrid S9.6 antibody[26] (ENH001, Kerafast) or with anti-rabbit IgG control antibody (#15410206, Diagenode). Complexes were captured with 50 µl DiaMag protein A- and protein G-coated magnetic beads (Diagenode, Seraing, Belgium) for 3 h at 4 °C. Subsequently, beads were washed three times in dilution buffer, incubated with RNase H (10 units, 60 min, 37 °C; NEB M0297L), washed again in dilution buffer and treated with 1× Proteinase K (Diagenode, Seraing, Belgium). RNA was extracted using QIAzol (QIAgen), chloroform and glycogen together with 5% input.

**Chromatin immunoprecipitation with DNA:RNA-hybrid antibodies**. To identify RNA-associated DNA fragments, human cardiomyocytes were crosslinked with UV light at 254 nm. Nuclei were isolated with the truCHIP™ Chromatin Shearing Kit (Covaris, USA) according to the manufacturer's protocol. Genomic DNA was fragmented using 25 mU/µl dsDNA Shearase Plus (Zymo research) for 5 min at 37 °C. After sonication with the Bioruptor Plus (10 cycles, 30 s on, 90 s off, 4 °C; Diagenode, Seraing, Belgium), cell debri was removed by centrifugation and dilution buffer (20 mM Tris-HCl, pH = 7.4; 100 mM NaCl; 2 mM EDTA; 0.5% Triton X-100; protease inhibitors) was added to lysates before pre-clearing with 20 µl DiaMag protein A- and protein G-coated magnetic beads (Diagenode, Seraing, Belgium) for 45 min at 4 °C. Five micrograms of anti-DNA-RNA Hybrid S9.6 antibody[26] (ENH001, Kerafast) or anti-rabbit IgG control antibody (#15410206, Diagenode) was added overnight at 4 °C. Complexes were captured with 50 µl DiaMag protein A- and protein G-coated magnetic beads (Diagenode, Seraing, Belgium) for 3 h at 4 °C, subsequently washed twice for 5 min with each of the wash buffers 1, 2, and 3 (1: 20 mM Tris-HCl, pH = 7.4; 150 mM NaCl; 0.1% SDS; 2 mM EDTA; 1% Triton X-100; 2: 20 mM Tris-HCl, pH = 7.4; 500 mM NaCl; 2 mM EDTA; 1% Triton X-100; 3: 10 mM Tris-HCl, pH = 7.4; 1% NP-40; 1% $C_{24}H_{39}NaO_4$; 1 mM EDTA). For RNase H samples[27], RNase H (10 units, 60 min, 37 °C; NEB M0297L) treatment was performed before washing with TE buffer, pH = 8.0. Elution was done with elution buffer (0.1 M NaHCO$_3$, 1% SDS) containing 1× Proteinase K (Diagenode, Seraing, Belgium) and shaking at 600 rpm for 1 h at 55 °C, 1 h at 62 °C and 10 min at 95 °C. After removal of the beads, the eluate was purified with the QiaQuick PCR purification kit (QIAgen, Hilden, Germany) together with 5% input and analyzed by qRT-PCR.

**Fractionation of nuclear and cytoplasmatic RNA**. One confluent T-75 flask of cells was washed with PBS, pelleted at 500 × *g* and 4 °C and lysed in 200 µl lysis buffer A (10 mM Tris (pH = 7.5), 10 mM NaCl, 3 mM MgCl$_2$ and 0.5% NP-40) for 5 min on ice. The supernatant was cleared by spinning at 1000 × *g* and 4 °C for 3 min and represented the cytoplasmatic fraction, to which QIAzol (QIAgen) was added for RNA isolation. The pellet was washed twice with lysis buffer A and lysed in 200 µl lysis buffer B (10 mM Tris (pH = 7.5), 150 mM NaCl, 3 mM MgCl$_2$) for

5 min on ice. After spinning at 1000 × *g* and 4 °C for 3 min, the pellet contained the nuclear fraction and was resuspended in QIAzol for phenol/chloroform RNA extraction.

**NMR spectroscopy**. DNA and RNA oligonucleotides were purchased from Sigma-Aldrich and 1 mM stocks of each strand were prepared by hydrating the oligonucleotides in 10 mM NaPO$_4$, 50 mM NaCl, 3 mM EGTA at pH = 7.4. NMR samples were 500 µl and contained 100 µM DNA duplex and contained 15 µl D$_2$O for a lock signal. Spectra were collected on an 800 MHz Bruker magnet equipped with a 5 mm CPTCI probe using a 1-1 jump and return pulse protocol. The pH of the NMR samples was measured using a Beckman Φ 340 pH meter and adjusted to 7.4 by addition of HCl. Equal molar ratios of RNA were added directly to the DNA duplex samples in the NMR tubes and mixed by inversion. Spectra were recorded at 288, 298 and 320 K with 288 K yielding the best signal dispersion. All spectra were processed using a sinc window multiplication function.

**CRISPR/Cas9-mediated excision *Sarrah* triple helix domain**. The mouse homolog of the triple helix domain in *Sarrah* (ENSMUST00000140003) together with 200 5′ and 3′ adjacently located nucleotides was used as input sequence for the RS2/Azimuth 2.0 gRNA-scoring algorithm[55]. Proposed gRNAs were selected based on their RS2 score and proximity to the triple helix forming domain. The lentiviral CRISPR/Cas9 gRNA-expressing vector pKLV2.2 (Addgene #72666)[56] was used to sequentially introduce two gRNAs (5′-TGTTGTATAATTCCCCTCAC-3′ and 5′-GAGTCCCAACAATTCCAGAA-3′) targeting the murine *Sarrah* triple helix domain. A vector containing control gRNAs was used as control. Positive cloning was sequentially confirmed by clonal SANGER sequencing. Lentiviruses were produced in HEK 293T cells using psPAX2 (Addgene #12260) as packaging plasmid and pMD2.G (Addgene #12259) as VSV-G envelope expressing plasmid[57]. Lentiviral transduction of HL-1 cells with pKLV2.2-constructs was performed together with a pLentiCRISPRv2_neo-construct (Addgene #98292) for 24 h. Cells were not selected resulting in a cell pool of wildtype cells and mutant cells lacking the triple helix forming domain of endogenously expressed *Sarrah*. Genotyping was performed using primers binding around the deleted region.

**RNA affinity purification**. HL-1 cells were lysed in lysis buffer (50 mM Tris-HCl, pH = 8; 150 mM NaCl; 1 mM EDTA; 1% NP-40; protease inhibitor) for 30 min on ice and volumes were adjusted to 1.1 ml with the same buffer lacking NP-40. For binding of RNA-protein complexes, lysates were pre-cleared for 2 h at 4 °C with blocked streptavidin C1 beads (Thermo Fisher; blocking with yeast tRNA and glycogen, both 0.2 mg/ml) and subsequently incubated with 200 pmol 2′O-Me-RNA oligonucleotides overnight at 4 °C. RNA-protein-oligonucleotide complexes were captured by addition of 100 µl blocked streptavidin C1 beads for 1 h at 37 °C. Beads were washed twice with wash buffer (50 mM Tris-HCl, pH = 8; 150 mM NaCl; 1 mM EDTA; 0.05% NP-40), twice with mild wash buffer (20 mM Tris-HCl, pH = 8; 10 mM NaCl; 1 mM EDTA; 0.05% NP-40) and once with mass spectrometry buffer (10 mM Tris-HCl, pH = 8; 50 mM NaCl). RNA and proteins were eluted by incubation at 95 °C for 5 min. Eluate and bead fractions were split for RNA extraction together with 10% input and mass spectrometry analysis. The portion for RNA extraction was resuspeded in QIAzol. DNase digestion was performed during RNA extraction.

**Mass spectrometry**. Proteins of eluted samples were separated by Bis-Tris SDS-PAGE (4–12% gradient gel, Novex, Life technologies). Proteins were reduced in 10 mM DTT, 50 mM ABC for one hour at 56 °C and alkylated for 45 min in 30 mM iodoacetamide. Samples were digested for 16 h with trypsin (sequencing grade, Promega) at 37 °C in 50 mM ABC, 0.01% Protease Max (Promega) and 1 mM CaCl$_2$. Liquid chromatography/mass spectrometry (LC/MS) was performed on Thermo Scientific™ Q Exactive Plus equipped with an ultra-high-performance liquid chromatography unit (Thermo Scientific Dionex Ultimate 3000) and a Nanospray Flex Ion-Source (Thermo Scientific). For data analysis MaxQuant 1.6.1.0[58], Perseus 1.6.1.3[59] and Excel (Microsoft Office 2013) were used.

**BrdU cell proliferation assay**. HUVECs were used for a BrdU flow cytometry-based cell proliferation assay after 48 h of co-cultivation with LNA GapmeR-treated human cardiomyocytes. The assay was performed using the BrdU Flow Kit (BD Biosciences) according to the manufacturer's instructions. Shortly, cells were incubated with BrdU for 45 min at 37 °C, washed with PBS, fixed with Cytofix/Cytoperm, permeabilized with Permeabilization Buffer Plus, re-fixed with Cytofix/Cytoperm and washed with PermWash buffer between all steps. Finally, cells were treated with DNase for 1 h at 37 °C to expose incorporated BrdU and stained with 2.5 µl of V450-anti-BrdU antibody (BD Biosciences #560810) as well as 20 µl of 7-AAD (BD Biosciences #559925) and analyzed with a BD FACSCanto II machine (BD Biosciences).

**In vitro sprouting assay**. Angiogenesis was modeled in vitro using a spheroid sprouting assay. For spheroid generation, 400 HUVECs were incubated in a

mixture of culture medium and methylcellulose (80%:20%) in a 96-well plate with non-adhesive U-bottoms to form spheroids. After 24 h at 37 °C, spheroids were collected by spinning at $200 \times g$ for 3 min, added to methylcellulose supplemented with FCS (80%:20%) and embedded in a collagen type-I gel (BD Biosciences). After gel polymerization, conditioned medium from GapmeR-treated human cardiomyocytes was added to the spheroids. 24 h later, gels were fixed with 4% formaldehyde in PBS and documented using an Axio Observer Z1.0 microscope (Zeiss) at fivefold magnification. Cumulative sprout length of each spheroid was measured using the AxioVision SE64 Rel. 4.9.1 software.

**Annexin V/7-AAD apoptosis assay.** Flow cytometry-based cell death assays were performed 24 or 48 h after transfection to measure early or late apoptosis, respectively. Cells were washed with PBS, pelleted at 4 °C together with dead cells from culture medium and resuspended in 100 μl Annexin V Binding Buffer (BD Biosciences). Five microliters of V450 Annexin V (BD Biosciences #560506) and 5 μl of 7-AAD were added and incubated for 15 min at room temperature. After addition of 200 μl Annexin V Binding Buffer, samples were analyzed with a BD FACSCanto II machine.

**Enzymatic SCOT1 activity assay.** HL-1 cell pellets were prepared 48 h after transfection by washing with PBS, pelleting at 4 °C and $800 \times g$ for 15 min and homogenizing for a succinyl-CoA:acetoacetate transferase (SCOT1) enzymatic activity assay. SCOT1 activity was measured as succinate-induced decrease in absorbance at 303 nm using a medium of the following composition: 100 mmol/l Tris-$H_2SO_4$ (pH = 8.05), 25 mmol/l $MgSO_4$, 50 μmol/l acetoacetyl-CoA (Sigma-Aldrich) and 0.1% (w/v) Triton X-100. Reactions were started by addition of sodium succinate at a final concentration of 50 mmol/l and absorbance at 303 nm was subsequently followed in time on a COBAS-FARA-centrifugal analyzer (Hoffmann-LaRoche).

**Statistical analysis.** Statistical analysis was performed using GraphPad Prism 8 software. Data are displayed as means ± SEM. For comparison between two normally distributed groups with normal data distribution, two-tailed paired or unpaired Student's $t$-tests was performed; for multiple comparisons, one-way analysis of variance (ANOVA) or two-way ANOVA followed by Bonferroni's correction was used. If normality of the data could not be confirmed, Mann–Whitney tests were used. Significant outliers within a group ($p < 0.05$) were detected by Grubbs's outlier test and excluded from the analysis. Data were considered statistically significant below a $p$-value of 0.05.

**Reporting summary.** Further information on research design is available in the Nature Research Reporting Summary linked to this article.

## Data availability

The datasets generated and analyzed during the current study are available from the corresponding author on reasonable request. Microarray and RNA sequencing data are deposited in the Gene Expression Omnibus repository under the accession number GSE145697 and GSE148146, respectively. Proteomics data are deposited in the PRIDE archive under the accession number PXD018315. Source data are provided as a Source Data file. To improve the transparency and the reproducibility of results a reporting summary is provided.

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

## Acknowledgements

We thank Denise Berghäuser, Silvia Rösser, Lorenz Pudelko, Simone-Franziska Glaser, Jana Meisterknecht, and Ruud Fontijn for technical assistance, Marion Muhly-Reinholz for support in histology, David John for help with bioinformatic analyses and Tamer Ali for help with the coding potential analysis. This work was supported by the German Centre for Cardiovascular Research (DZHK), the Cardiopulmonary Institute (CPI), the Deutsche Forschungsgemeinschaft (SFB834 to R.A.B. and S.D., SFB815 to I.W., TRR267 to M.S.L., R.P.B., S.E., N.J., I.W., J.K., S.D., and R.A.B.), the European Research Council ("NOVA" to R.A.B. and "AngioInc" to S.D.), the Netherlands Organisation for Scientific Research (NWO Vidi to R.A.B.), the European Union (Horizon 2020 grant no. 825670 to R.A.B.), the NSF (0922862 to C.N.J), NIH (S10 RR025677 to C.N.J) and Vanderbilt University matching funds (to C.N.J).

## Author contributions

D.J.T. designed and performed experiments, analyzed data and drafted the article; D.I.B., K.T., J.S., A.F., A.v.B., C.S., M.S.L., R.P.B., T.A., B.B., A.W., C.N.J., A.S.-C., M.K., S.W., M.N.H., K.Y., L.K., P.H., I.W., N.H., C.B., J.K., and R.H.H. performed experiments; C.K. and I.G.C. performed computational analyses; S.E. and T.E. critically revised the manuscript; N.J. provided technical and conceptual advice; S.D. provided conceptual input and critically revised the manuscript; R.A.B. oversaw the project, designed experiments, analyzed data, and drafted the article.

## Competing interests

D.J.T., R.A.B., and S.D. have filed a patent about the therapeutic use of the lncRNA *Sarrah*. The remaining authors declare no competing interests.
