## [Peer Review File · Nature Communications]

Reviewers' comments:

Reviewer #1 (Remarks to the Author):

Trembinski et al. Sarrah is an Aging-Regulated Anti-Apoptotic Long Non-Coding RNA in Cardiomyocytes that Augments Recovery from Acute Myocardial Infarction

This manuscript deals with the discovery of an age-regulated long non-coding RNA termed “Sarrah” and deciphers its location, differential expression and function in the context of the senescent heart and in acute myocardial infarction. The study contains a plethora of techniques and models, but could benefit from readjustments in the study design and more focus to justify the claims and conclusions about this lncRNA.

Major

1. Fig.1c: The description of Sarrah expression in cardiac cells is too vague with a separation between cardiomyocytes versus stromal cells. A more informative separation minimally includes cardiomyocytes, fibroblasts/pericytes and endothelial cells and qRT-PCR or Taqman analyses.
2. Fig.1e: in the context of this manuscript, it is unclear why the authors studied hypertensive rats, HFpEF rats and rats with HFpEF and high fat diet. The added value of these disease subtypes distracts from the main conclusion that Sarrah is age-related and influences apoptotic pathways, and raises the question if Sarrah functionally participates in the HFpEF syndrome, where apoptosis is seemingly of a minor role in the pathogenesis.
3. In Fig.2, the authors quickly continue with an AAV gene delivery study for Sarrah in the setting of acute myocardial infarction in young mice, which is incongruent with its reported function in age-related heart disease. A far more impressive approach would be to rescue the hearts from a (prematurely) aged mouse model. A therapeutic study aimed to resupplement Sarrah expression would be expected later in the manuscript.

4. A more vexing question is whether Sarrah downregulation (either by Gapmers or using the Cisprr model described later in the manuscript) is sufficient to evoke excessive apoptosis in wildtype mice and contributes to signs of cardiac senescence and/or apoptosis. If no additional signs of senescence are found, then the original term “age-related” was possibly prematurely chosen (this could be counter argued by selecting a different mouse model, see comment #3 above), and suggests Sarrah is only involved in cellular death pathways.

5. More specifically on the execution of mentioned study in Fig.1, can the authors confirm that Sarrah overexpression was restricted to the heart and if so, only to heart muscle cells (Fig.1d)? If so, the endothelial component (Fig.1i) can only indirectly be affected in this study. The rescue in LVEF was relatively mild (Fig.1e), even when data are expressed as delta values. A more accurate physiological assessment could have been derived from MRI studies, invasive hemodynamics and/or including Doppler assessments. As presented, parameters of wall thickness and movements and ventricular volumes remain obscure. The quantification of the infarct size is suboptimal and should be derived from a 3D geometry of serially sectioned hearts. Fig.1g, h: quantification of TUNEL positive events and proliferating cells should be expressed per number of nuclei, not mm² of histological sections. By what mechanism does Sarrah influence cardiomyocyte proliferation?

6. In the co-culture experiments with cardiomyocytes, a convincing approach would be to overexpress Sarrah and observe an increase in the proliferative capacity of HUVECs in the EHT-organoids. Do the authors think that Sarrah has a cell autonomous effect in endothelial cells or is the prevailing mechanism a form of communication between endothelial cells and cardiomyocytes, in which case conditioned media and the exosomal content could be of interest to study.

7. Fig.4: Some level of experimental validation of Sarrah downstream targets and their involvement in apoptosis are warranted.

Minor

1. Write the name of the lncRNA in italics
2. Lines 94-95: rephrase to make clearer which of the two fractions (non-/cardiomyocytes) or both have been used for the RNA isolation;
3. Line 112: “ontogenic” instead of “ontogenous”;
4. Figure 1a: specify that the red dot corresponds to Sarrah-siRNA;

5. Figure 2f: change “rest” with “non-cardiomyocytes”.

Reviewer #2 (Remarks to the Author):

The paper by Trembinski et al. evaluates the cardiac lncRNA transcriptome during aging. Via comparing RNA samples from young vs. old mouse heart cardiomyocytes, the authors identified >5000 expressed lncRNAs, of which 76 were selected for further validation. A siRNA screen, in combination with an apoptotic cell-based assay, finally identified Sarrah, a lncRNA located in the *Oxct1* gene locus. Specifically, Sarrah knockdown increases apoptosis in cardiomyocytes. Next, the authors investigate the role of Sarrah in vivo in a mouse model of myocardial infarction, and demonstrate that Sarrah overexpression is protective in the stressed cardiomyocyte population. Several other experimental conditions are studied and confirm detrimental effects of Sarrah silencing on cardiomyocyte behavior. The proposed molecular mechanisms associated with Sarrah action could include formation of triple RNA-DNA helices at target gene promoters.

This paper is interesting in its initial approach to identify aged-related lncRNAs in cardiomyocytes. The study is however quite confusing on several aspects. Sarrah is identified as expressed in the aged mouse cardiomyocyte population, is associated with apoptosis in the HL-1 cell line, and finally is demonstrated to produce beneficial effects in cardiomyocytes after myocardial infarction in vivo. These beneficial effects result from a role of Sarrah in apoptosis, in cardiomyocyte contractility, in regulating fibrosis and possibly in controlling heart perfusion via affecting endothelial cell function in the damaged heart. In addition, the authors take advantage in their study of mouse models of aging and myocardial infarction, of a rat model of HFpEF, and then of mouse HL-1 cells as well as human and rat cardiomyocytes in vitro. At the end, if Sarrah appears to be protective in all these different instances/pathophysiological situations, its exact role is not determined. Moreover, the molecular mechanisms responsible for these pleiotropic effects are not identified. If cardiomyocyte apoptosis is postulated to be at the center of Sarrah action, this should be studied in more details and the molecular mechanisms should be clearly described.

Major points

1. It is not clear how was Sarrah identified. The authors initially identified 5439 annotated lncRNAs as expressed in the young and old mouse hearts. How many lncRNAs among these 5439 were differentially expressed between the young and aged hearts? How were the 76 selected lncRNAs chosen for further study? Were these 76 lncRNAs differentially expressed? Please provide the list of the 76 genes together with the list of the adjacent protein coding genes (PCGs). Were these PCGs related to apoptosis, other pathways?

2. Sarrah expression does not seem to be significantly downregulated in old mouse cardiomyocytes (Figure 1C). Sarrah is also not changed in old vs. young non-myocyte cells. However, Sarrah is significantly downregulated in the old mouse heart as compared to the young mouse heart (Figure 1D). It doesn't seem to add up with the assumption that Sarrah is predominantly expressed in cardiomyocytes in vivo.

3. Sarrah knockdown in cardiomyocytes affects their contractility. How can the authors exclude that this effect does not occur secondary of induced apoptosis? What is the postulated mechanism linking Sarrah and contractility? Do some Sarrah target genes encode sarcomeric proteins, calcium handling proteins, etc.

4. Sarrah is presented as a conserved transcript but data supporting this statement are not shown. Please provide evidence for sequence/syntenic conservation between mouse, rat and human Sarrah. Figure 1B, as mentioned in page 6, bottom, does not provide this information.

5. Moreover, Sarrah is presented as a hypoxia-responsive gene. What is the evidence for this?

6. The formation of triple helices at target gene promoters is proposed to play a role in Sarrah action. The authors mention enrichment of triple helix prone motifs at promoters of genes modulated after Sarrah silencing in human cardiomyocytes as assessed by using the Triplex Domain Finder software. This should be shown and the list of genes should be provided.

7. The Triplex Domain Finder analysis is performed presumably using the human sequence (please confirmed). How conserved is this putative mechanism? The authors mention little conservation in Sarrah sequences from different species. Does mouse Sarrah for instance contain a region predicted to form triple helices? What are the sequences that are deleted in mouse Sarrah in experiments depicted in Figure 4H.

8. Why considering only downregulated genes after Sarrah silencing for the Triplex Domain Finder analysis (Suppl. Figure 6A)? Sarrah might exert repressive action on target genes, which could be alleviated following Sarrah knockdown. Upregulated genes should not be excluded a priori of the analysis. In any case, it would be important to compare up and downregulated genes, and to determine whether a disequilibrium is observed between these two gene sets in terms of percentage of genes containing triple helix prone motifs.

9. Is there overlap in the genes, which are downregulated following Sarrah silencing in human vs. mouse cardiomyocytes, and which contain a triple helix prone motif in their promoters?

10. Sarrah seems to exert trans-regulatory function on gene expression. The proposed mechanism via formation of triple helices is interesting but has not been previously described. Instead, lncRNAs have been extensively described as being able to partner with proteins, in particular chromatin modifiers. This mechanism seems to be quite relevant in the context of the present study. It is not clear therefore why the authors exclude this possibility. The authors could consider a RNA pulldown approach to identify Sarrah-bound protein. Along this vein, deletion in Sarrah sequences might as well alter binding of important protein partners.

11. Figure 4I. Overexpression of mouse Sarrah lacking the triple helix domain increases apoptosis in mouse cardiomyocytes. This is surprising. Overexpression of WT Sarrah should decrease apoptosis

(This control should be included in the experiment), and possibly delta Sarrah should not produce any effects on cardiomyocyte apoptosis but there are no obvious reasons for the deleted transcript to increase apoptosis.

12. The target loci of Sarrah action should be investigated using a ChIRP approach.

Minor points

1. Often, the legends, and even the methods are not clear. For instance, given the variety of models used in the paper, the authors should carefully indicate in the legend which cell types and which species are studied in the various experimental approaches described in the paper. In particular, the authors should clearly indicate when a cell line is used and when they rely on primary cells. Several key methods are not described, for instance the Triplex Domain Finder analysis.

2. The protocol using total Histone 3 immunoprecipitation to detect potential association of Sarrah with chromatin (Fig. 4C) should include a step in which precipitated material is treated with DNase to avoid possible amplification of Sarrah-encoding DNA.

Point-By-Point Reply to the Reviewers

Manuscript:

Sarrah is an Aging-Regulated Anti-Apoptotic Long Non-Coding RNA in Cardiomyocytes that Augments Recovery from Acute Myocardial Infarction; Trembinski et al.

First of all, we would like to thank the reviewers for their very helpful comments and overall their efforts, which helped us to substantially improve the manuscript and to more thoroughly describe the role of lncRNA *Sarrah* in the heart.

Response to reviewer #1

This manuscript deals with the discovery of an age-regulated long non-coding RNA termed “*Sarrah*” and deciphers its location, differential expression and function in the context of the senescent heart and in acute myocardial infarction. The study contains a plethora of techniques and models, but could benefit from readjustments in the study design and more focus to justify the claims and conclusions about this lncRNA.

Major

1. Fig. 1c: The description of *Sarrah* expression in cardiac cells is too vague with a separation between cardiomyocytes versus stromal cells. A more informative separation minimally includes cardiomyocytes, fibroblasts/pericytes and endothelial cells and qRT-PCR or Taqman analyses.

We thank the reviewer for this helpful comment and agree that a separation of stromal cells into fibroblasts, endothelial cells and leukocytes would be more informative. We have therefore isolated cardiomyocytes, fibroblasts and endothelial from mouse hearts and measured *Sarrah* levels in all fractions (reviewer fig. 1, fig. 1c, supplementary fig. 1e). To condense the paper, we have moved the original figure to the supplements. The results indicate that *Sarrah* is expressed at comparable levels in endothelial cells and cardiomyocytes, but lower in fibroblasts. In accordance with these new findings, we removed the term “cardiomyocyte-enriched” from the manuscript. In total heart lysates, in which the majority of the RNA comes from cardiomyocytes, *Sarrah* is significantly reduced with aging (fig. 1d).

Reviewer fig. 1: 12 week old mice were used for cell isolation of cardiomyocytes (CM), endothelial cells (EC) and fibroblasts (FB)). RNA was isolated and levels of cell type-specific markers (Tnnt2 for CM, Cdh5 for EC) and *Sarrah* were determined by qRT-PCR.

2. Fig. 1e: In the context of this manuscript, it is unclear why the authors studied hypertensive rats, HFpEF rats and rats with HFpEF and high fat diet. The added value of these disease subtypes distracts from the main conclusion that *Sarrah* is age-related and influences apoptotic pathways, and raises the question if *Sarrah* functionally participates in the HFpEF syndrome, where apoptosis is seemingly of a minor role in the pathogenesis.

The reviewer made a valid point here. We included the data derived from a HFpEF rat model to emphasize a potential of *Sarrah* during aging as HFpEF and cardiomyocyte stiffness mainly occur in the elderly. We agree though that the data might distract the reader from the main story line of the manuscript, which is about anti-apoptotic effects of *Sarrah* during aging and after acute myocardial infarction (AMI), and therefore moved these data to the supplementary figures (supplementary fig. 1g).

3. In Fig. 2, the authors quickly continue with an AAV gene delivery study for *Sarrah* in the setting of acute myocardial infarction in young mice, which is incongruent with its reported function in age-related heart disease. A far more impressive approach would be to rescue the hearts from a (prematurely) aged mouse model. A therapeutic study aimed to resupplement *Sarrah* expression would be expected later in the manuscript.

The reviewer made a very interesting point here. In the six month revision period, we did not have enough time to apply for ethical permission to use a progeria model (our choice would have been Ku80^{-/-} mice), obtain these mice, breed enough numbers and perform experiments with 8-week-long overexpression of *Sarrah*. However, we have now included more mice in the AAV gene delivery study and additionally performed MRI analysis of cardiac function. Of note, the mice in the study are subjected to acute myocardial infarction, which is age-related and of particular therapeutic relevance. Furthermore, we moved the *in vivo* part of the manuscript towards the end (new fig. 5), as the reviewer suggests.

4. A more vexing question is whether *Sarrah* downregulation (either by Gapmers or using the Crispr model described later in the manuscript) is sufficient to evoke excessive apoptosis in wildtype mice and contributes to signs of cardiac senescence and/or apoptosis. If no additional signs of senescence are found, then the original term “age-related” was possibly prematurely chosen (this could be counter argued by selecting a different mouse model, see comment #3 above), and suggests *Sarrah* is only involved in cellular death pathways.

We thank the reviewer for the suggestion and share the notion that it would be indeed interesting to study the effects of *Sarrah* knockdown or knockout *in vivo*.

Therefore, we intraperitoneally injected GapmeRs into 12 week old male C57Bl6/J mice to obtain an *in vivo* *Sarrah* knockdown model (reviewer fig. 2). We first injected the GapmeR sequence we used for our *in vitro* studies (TTGGAAAGGTGAGCTG) at 20 mg/kg, but obtained only a slight knockdown in the heart atrium and no knockdown in the ventricle. Injecting three doses of the GapmeR did not improve the knockdown efficiency. Eventually, we injected another GapmeR sequence (CAACATGGACAAGGAC) at 20 mg/kg, but still did not achieve a reasonable knockdown in the mouse heart.

Reviewer fig. 2: *Sarrah* levels in the heart atrium and ventricle, liver and skeletal muscle following different GapmeR injection protocols.

We apologize that it has not been feasible to generate a knockdown model for *Sarrah* and hope that the reviewers will appreciate our efforts. Furthermore, we find it more therapeutically relevant to show that *Sarrah* augmentation induces cardiac function, which we confirmed in a second experiment (see also below). The reviewer states that the term “age-related” is prematurely chosen. We used the term “age-regulated” merely to denote that aging reduces the expression of *Sarrah*.

5. More specifically on the execution of mentioned study in Fig. 1, can the authors confirm that *Sarrah* overexpression was restricted to the heart and if so, only to heart muscle cells (Fig. 1d)? If so, the endothelial component (Fig. 1i) can only indirectly be affected in this study. The rescue in LVEF was relatively mild (Fig. 1e), even when data are expressed as delta values. A more accurate physiological assessment could have been derived from MRI studies, invasive hemodynamics and/or including Doppler assessments. As presented, parameters of wall thickness and movements and ventricular volumes remain obscure. The quantification of the infarct size is suboptimal and should be derived from a 3D geometry of serially sectioned hearts. Fig. 1g, h: quantification of TUNEL positive events and proliferating cells should be expressed per number of nuclei, not mm² of histological sections. By what mechanism does *Sarrah* influence cardiomyocyte proliferation?

These are all very valid points and we have addressed them accordingly. Firstly, *Sarrah* overexpression was performed using well characterized AAV9, which have been repeatedly used for cardiomyocyte gene transfer. These vectors only transduce cardiomyocytes and do not target endothelial cells or fibroblasts adequately at the dose used in our study (6×10^{11} viral genomes per mouse; [1]). Indeed the effects on endothelial cell proliferation are very likely paracrine. Co-culture experiments *in vitro* confirm that *Sarrah* depletion in cardiomyocytes reduces proliferation of endothelial cells (supplementary fig 7c). Conversely, overexpression of *Sarrah* in cardiomyocytes induces endothelial cell proliferation (supplementary fig. 7b and reviewer fig. 6).

Secondly, at the request of the reviewer we have repeated the AMI experiment with GFP-control and *Sarrah* overexpressing mice, using the exact conditions as originally performed. Now we included cardiac MRI measurements to quantify cardiac function (fig. 5c-h, supplementary fig. 6e, g, h and reviewer fig. 4). This experiment showed that our echocardiography analysis of the original study was very comparable to MRI measurements. Furthermore, we now include additional parameters in our main figures (fig. 5c-h) and supplemental figures (supplementary fig. e, g, h). At the end of this experiment, we used the hearts to perform serial sectioning as requested by the reviewer (fig. 5l and reviewer fig. 5).

Reviewer figure 4:

- Adeno-associated virus serotype 9 (AAV9)-green fluorescent protein (GFP) control virus particles or AAV9-cytomegalovirus (CMV)-*Sarrah* virus particles were injected intravenously into mice two weeks prior to AMI surgery. Cardiac function was analyzed by echocardiography at 1, 7 and 14 days and by MRI at 14 days after AMI surgery.
- Sarrah* levels of mouse hearts after AMI were measured by qRT-PCR. (n = 15-16; SEM; * t-test p < 0.05).
- Wall motion score index (WMSI), a parameter of heart muscle movement at 16 individual heart sections, was assessed from echocardiographic measurements. A WMSI decrease indicates an improved recovery of cardiac contractile function. Delta WMSI values at day 14 refer to day 1 (n = 6-9; SEM; * t-test p < 0.05).
- Left ventricular wall thickness was assessed from magnetic resonance imaging (MRI) at day 14 after AMI surgery (n = 7-8; SEM; * t-test p < 0.05).
- Stroke volume was calculated as the difference between end diastolic and end systolic volumes, which were assessed from MRI at day 14 after AMI surgery (n = 7-8; SEM; ** t-test p < 0.01).
- Cardiac contractile function was assessed by MRI. Displayed are delta ejection fraction values on day 14 after AMI in comparison to day 1 (n = 7-8; SEM; ** t-test p < 0.01).
- Representative images from MRI on day 14 after AMI during end systole and end diastole from apical to basal heart segments.

Reviewer figure 5: Serial sections of AMI hearts were stained using sirius red and infarct size was assessed by circumference of the infarcted region as percentage of the left ventricle (n = 7-8; SEM; * t-test p < 0.05; representative images of one animal per group are shown; scale bars are 1 mm).

Lastly, we have quantified apoptosis and proliferation normalized to number of nuclei (fig. 5i-j). Our data indicate that cardiomyocyte proliferation is not affected by *Sarrah*. *Sarrah* overexpression does induce proliferation of endothelial cells in a paracrine manner (see also next point).

6. In the co-culture experiments with cardiomyocytes, a convincing approach would be to overexpress *Sarrah* and observe an increase in the proliferative capacity of HUVECs in the EHT-organoids. Do the authors think that *Sarrah* has a cell autonomous effect in endothelial cells or is the prevailing mechanism a form of communication between endothelial cells and cardiomyocytes, in which case conditioned media and the exosomal content could be of interest to study.

Thank you for this interesting comment. We agree that an overexpression approach would be more convincing and therefore overexpressed *Sarrah* in human cardiomyocytes, co-cultured them with HUVECs for 48 hours and analyzed HUVEC proliferation by BrdU staining followed by flow cytometry (supplementary fig. 7b and reviewer fig. 6). Indeed, after co-culture with *Sarrah* overexpressing cardiomyocytes, more HUVECs in S-phase are observed. Since we now focus the manuscript on the effects of *Sarrah* on cardiomyocyte apoptosis, we have refrained from describing the effects of cardiomyocyte *Sarrah* on endothelial cells in more detail in the manuscript.

Reviewer figure 6: Human cardiomyocytes were transduced with lentiviral vectors for overexpression and co-cultured with HUVECs on a transwell after medium change. Proliferation of HUVECs was assessed by bromodeoxyuridine (BrdU) staining and flow cytometry analysis after two days of co-culturing (n = 3; SEM; * t-test p < 0.05).

7. Fig. 4: Some level of experimental validation of *Sarrah* downstream targets and their involvement in apoptosis are warranted.

We have identified 135 downstream target genes. We selected the 5 that we hypothesized to be most likely to confer the anti-apoptotic effects of *Sarrah*. However, knockdown of these 5 genes (GPC6, PDE3A, ITPR2, PARP8, SSBP2) did not significantly induce apoptosis (data not shown). We hypothesize that, in analogy to microRNA function, regulation of the majority of targets together confers the anti-apoptotic action of *Sarrah*.

Minor

1. Write the name of the lncRNA in italics.
2. Lines 94-95: Rephrase to make clearer which of the two fractions (non-/cardiomyocytes) or both have been used for the RNA isolation.
3. Line 112: "ontogenic" instead of "ontogenous".
4. Figure 1a: specify that the red dot corresponds to *Sarrah*-siRNA.
5. Figure 2h: change "rest" with "non-cardiomyocytes".

We thank the reviewer for his or her attention to detail. We have implemented the changes.

Response to reviewer #2:

The paper by Trembinski et al. evaluates the cardiac lncRNA transcriptome during aging. Via comparing RNA samples from young vs. old mouse heart cardiomyocytes, the authors identified >5000 expressed lncRNAs, of which 76 were selected for further validation. A siRNA screen, in combination with an apoptotic cell-based assay, finally identified *Sarrah*, a lncRNA located in the *Oxct1* gene locus. Specifically, *Sarrah* knockdown increases apoptosis in cardiomyocytes. Next, the authors investigate the role of *Sarrah* in vivo in a mouse model of myocardial infarction, and demonstrate that *Sarrah* overexpression is protective in the stressed cardiomyocyte population. Several other experimental conditions are studied and confirm detrimental effects of *Sarrah* silencing on cardiomyocyte behavior. The proposed molecular mechanisms associated with *Sarrah* action could include formation of triple RNA-DNA helices at target gene promoters.

This paper is interesting in its initial approach to identify aged-related lncRNAs in cardiomyocytes. The study is however quite confusing on several aspects. *Sarrah* is identified as expressed in the aged mouse cardiomyocyte population, is associated with apoptosis in the HL-1 cell line, and finally is demonstrated to produce beneficial effects in cardiomyocytes after myocardial infarction in vivo. These beneficial effects result from a role of *Sarrah* in apoptosis, in cardiomyocyte contractility, in regulating fibrosis and possibly in controlling heart perfusion via affecting endothelial cell function in the damaged heart. In addition, the authors take advantage in their study of mouse models of aging and myocardial infarction, of a rat model of HFpEF, and then of mouse HL-1 cells as well as human and rat cardiomyocytes in vitro. At the end, if *Sarrah* appears to be protective in all these different instances/pathophysiological situations, its exact role is not determined. Moreover, the molecular mechanisms responsible for these pleiotropic effects are not identified. If cardiomyocyte apoptosis is postulated to be at the center of *Sarrah* action, this should be studied in more details and the molecular mechanisms should be clearly described.

Major

1. It is not clear how was *Sarrah* identified. The authors initially identified 5439 annotated lncRNAs as expressed in the young and old mouse hearts. How many lncRNAs among these 5439 were differentially expressed between the young and aged hearts? How were the 76 selected lncRNAs chosen for further study? Were these 76 lncRNAs differentially expressed? Please provide the list of the 76 genes together with the list of the adjacent protein coding genes (PCGs). Were these PCGs related to apoptosis, other pathways?

Considering the comparison of aged cardiomyocyte fraction with young cardiomyocyte fraction, there are 91 differentially regulated lncRNAs ($p < 0.05$). We manually curated the list of cardiomyocyte-enriched lncRNAs and removed lncRNAs for which we did not find reliable reads when assessing the data in a genome viewer. At the request of the reviewer, we have prepared the list of 76 lncRNAs with adjacent protein-coding genes (PCGs). Of all PCGs on both sides of the lncRNAs we performed a Pubmed search to assess a potential role in apoptosis (see Excel file “76 lncRNA and PCGs”). However, we did not select *Sarrah* based on the genomic location. We used this list of 76 lncRNAs as a starting point for a functional assay to identify lncRNAs that regulate apoptosis in cardiomyocytes. This assay showed that out of these 76 lncRNAs, the most robust induction of apoptosis was achieved by silencing *Sarrah*.

2. *Sarrah* expression does not seem to be significantly downregulated in old mouse cardiomyocytes (Figure 1C). *Sarrah* is also not changed in old vs. young non-myocyte cells. However, *Sarrah* is significantly downregulated in the old mouse heart as compared to the young mouse heart (Figure 1D). It doesn't seem to add up with the assumption that *Sarrah* is predominantly expressed in cardiomyocytes in vivo.

We thank the reviewer for the valid remark. To address this concern, we have isolated cardiomyocytes (CM), endothelial cells (EC) and fibroblasts (FB) from mouse hearts and measured levels of cell-specific markers and *Sarrah* (see also point #1 from reviewer #1). The results indicate that *Sarrah* is expressed at comparable levels in endothelial cells and cardiomyocytes, but lower in fibroblasts. In accordance with these new findings, we removed the term “cardiomyocyte-enriched” from the manuscript. In total heart lysates, in which the majority of the RNA comes from cardiomyocytes, *Sarrah* is significantly reduced with aging (reviewer fig. 7, fig. 1c, supplementary fig. 1e).

Reviewer figure 7: 12 week old mice were used for cell isolation of cardiomyocytes (CM), endothelial cells (EC) and fibroblasts (FB)). RNA was isolated and levels of cell type-specific markers (Tnnt2 for CM, Cdh5 for EC) and *Sarrah* were determined by qRT-PCR.

3. *Sarrah* knockdown in cardiomyocytes affects their contractility. How can the authors exclude that

this effect does not occur secondary of induced apoptosis? What is the postulated mechanism linking *Sarrah* and contractility? Do some *Sarrah* target genes encode sarcomeric proteins, calcium handling proteins, etc.

The rat cardiomyocyte contractility measurements were performed on individual viable beating cardiomyocytes, therefore we already select for cardiomyocytes that have not undergone apoptosis yet. Several *Sarrah* target genes indeed regulate contractility. For example phosphodiesterase 3a (encoded by the *Pde3a* gene) is known to regulate Ca^{2+} handling via regulation of PKA signaling. We postulate that *in vivo*, the increased cardiac contractile function after *Sarrah* overexpression is achieved through a combination of the following mechanisms: (1) increased survival of cardiomyocytes, leading to a higher number of cells that contribute to contractile function, (2) increased contractile function of individual cardiomyocytes and (3) paracrine stimulation of endothelial cell proliferation likely contributing to increased perfusion.

4. *Sarrah* is presented as a conserved transcript but data supporting this statement are not shown. Please provide evidence for sequence/syntenic conservation between mouse, rat and human *Sarrah*. Figure 1B, as mentioned in page 6, bottom, does not provided this information.

We apologize for the lack of clarity. Unlike protein coding genes, most lncRNA genes are poorly sequence-conserved, but rather locus-conserved [2]. Indeed *Sarrah* is locus-conserved, which we mention in the manuscript: “we searched for homologous transcripts in humans, pigs and rats using publicly available sequencing and annotation databases (<http://genome.ucsc.edu/>; 18–22) and found transcripts in the ontogenic loci with small stretches of conserved sequences” (fig. 1b, supplementary fig. 1d, supplementary table 1a). Nevertheless, *Sarrah* sequence conservation between mouse and human is relatively high (supplementary fig. 1d and reviewer fig. 8), which provides further evidence for its conserved function and relevance.

Reviewer figure 8: The sequences of human and mouse *Sarrah* are partially conserved.

5. Moreover, *Sarrah* is presented as a hypoxia-responsive gene. What is the evidence for this?

We have investigated whether *Sarrah* is regulated in a hypoxia-responsive manner in both mouse and human cardiomyocytes by culturing the cells at 1 % and 0.2 % oxygen, respectively (supplementary fig. 6a-b and reviewer fig. 9). Additionally, we treated the cells with 300 μ M deferoxamine mesylate (DFO), a hypoxia-mimicking compound. All conditions yielded a significant reduction in *Sarrah* levels and have led to the conclusion that *Sarrah* is downregulated by hypoxia.

Reviewer figure 9: *Sarrah* is downregulated by hypoxia and DFO treatment *in vivo*.

- a. Hypoxia was induced in mouse (HL-1 cell line) cardiomyocytes by treatment with 300 μ M *deferioxamine mesylate* (DFO), a chelating compound, or by exposure to hypoxia (1 % O₂). VEGFA upregulation was measured by qRT-PCR as a hypoxic marker; *Sarrah* levels were measured by qRT-PCR (n = 3; SEM; * t-test p < 0.05; ** t-test p < 0.01).
- b. Hypoxia was induced in primary human cardiomyocytes by treatment with 300 μ M *deferioxamine mesylate* (DFO), a chelating compound, or by exposure to hypoxia (0.2 % O₂). VEGFA upregulation was measured by qRT-PCR as a hypoxic marker; *Sarrah* levels were measured by qRT-PCR (n = 3; SEM; * t-test p < 0.05; ** t-test p < 0.01; *** t-test p < 0.001).

6. The formation of triple helices at target gene promoters is proposed to play a role in *Sarrah* action. The authors mention enrichment of triple helix prone motifs at promoters of genes modulated after *Sarrah* silencing in human cardiomyocytes as assessed by using the Triplex Domain Finder software. This should be shown and the list of genes should be provided.

We have specified the regions in mouse and human *Sarrah* forming triple helices for the reviewers (see supplementary table 3). Additionally, we have prepared a list with the complete results from the Triplex Domain Finder software which includes all target genes and exact RNA binding sites in promoter regions of target genes for both human and mouse *Sarrah* (see Excel file “Triplex domain finder_*Sarrah* results”).

7. The Triplex Domain Finder analysis is performed presumably using the human sequence (please confirmed). How conserved is this putative mechanism? The authors mention little conservation in *Sarrah* sequences from different species. Does mouse *Sarrah* for instance contain a region predicted to form triple helices? What are the sequences that are deleted in mouse *Sarrah* in experiments depicted in Figure 4H?

We confirm that the Triplex Domain Finder analysis was initially performed with human data, which identified the triple helix domain of *Sarrah* and 135 target genes with triple helix prone motifs in the human genome, all of them being downregulated in a microarray experiment with *Sarrah*-depleted human cardiomyocytes.

Subsequently, we repeated the analysis using mouse *Sarrah* and promoter sequences from the mouse genome. As in human, the algorithm detected a triple helix forming motif in the murine

Sarrah sequence as well as an enrichment of triple helix prone motifs in downregulated, but not upregulated gene promoters (see next point for enrichment in up- vs. downregulated genes in human).

The sequence deleted in mouse cardiomyocytes (HL-1 cell line) in figure 3h and supplementary figures 5e-g is the triple helix forming region in mouse (sequences of sgRNAs used in the experiment are provided in supplementary table 5c).

Sarrah target genes showed a significant overlap between human and mouse (see concern #9; list provided, sheet “Overlapping *Sarrah* targets” in Excel file “Triplex domain finder_*Sarrah* results”).

8. Why considering only downregulated genes after *Sarrah* silencing for the Triplex Domain Finder analysis (Suppl. Figure 6A)? *Sarrah* might exert repressive action on target genes, which could be alleviated following *Sarrah* knockdown. Upregulated genes should not be excluded a priori of the analysis. In any case, it would be important to compare up- and downregulated genes, and to determine whether a disequilibrium is observed between these two gene sets in terms of percentage of genes containing triple helix prone motifs.

We apologize for not describing the analysis clear enough in the original manuscript. Our analysis considered both up- and downregulated genes. However, the Triplex Domain Finder statistical analysis identified an enrichment of triple helix formation only between *Sarrah* and downregulated genes (human: $p = 0.0033$; mouse: $p = 0.0012$), while no enrichment was detected for upregulated genes (human: $p = 0.31$; mouse: $p = 0.99$). Similar results were obtained with mouse sequences. With regards to percentages: 27 % of downregulated human genes and 33 % of downregulated mouse genes contain triple helix motifs.

9. Is there overlap in the genes, which are downregulated following *Sarrah* silencing in human vs. mouse cardiomyocytes, and which contain a triple helix prone motif in their promoters?

This is indeed an interesting and relevant question. We identified 165 mouse genes that were predicted to contain triple helix motifs for mouse *Sarrah* in their promoters. Out of the 135 human gene promoters forming triple helices with *Sarrah*, 54 (40 %) were also targeted by mouse *Sarrah*. This includes for example the promoter of *GPC6* that we characterized in both species. The list of overlapping target genes is provided for the reviewers (sheet “Overlapping *Sarrah* targets” in Excel file “Triplex domain finder_*Sarrah* results”). Of note, even though a 40 % overlap may seem low, this is in the same order of magnitude as target genes of well-conserved microRNAs. For example, the overlap of miR-34a targets between mouse and human is 54 % (according to Targetscan v7.2).

10. *Sarrah* seems to exert trans-regulatory function on gene expression. The proposed mechanism via formation of triple helices is interesting but has not been previously described. Instead, lncRNAs have been extensively described as being able to partner with proteins, in particular chromatin modifiers. This mechanism seems to be quite relevant in the context of the present study. It is not clear therefore why the authors exclude this possibility. The authors could consider a RNA pulldown approach to identify *Sarrah*-bound protein. Along this vein, deletion in *Sarrah* sequences might as well alter binding of important protein partners.

We fully agree that an RNA pulldown approach would be a highly relevant means for further characterizing the mechanism of *Sarrah* function and have addressed the reviewer's concern as suggested. RNA pulldown of *Sarrah* from lysate of the mouse cardiomyocyte cell line HL-1 followed by mass spectrometry identified CRIP2, a transcription factor expressed mainly in the heart, as an interaction partner of *Sarrah* (fig. 4b-c and reviewer fig. 10a-b), which is known to recruit the transcriptional co-activator p300 [3] that acetylates histone H3 on lysine 27 (H3K27ac) to activate transcription. We validated the physical interaction between CRIP2 and *Sarrah*, between p300 and *Sarrah* as well as between H3K27ac and *Sarrah* in RIP experiments (fig. 5d and reviewer fig. 10c) and conclude that *Sarrah* facilitates gene transcription through recruiting CRIP2 and p300 to open chromatin loci. Of note, triple helix-mediated RNA-DNA interaction in *cis* has been shown for several lncRNAs (see <http://dx.doi.org/10.1016/j.chembiol.2016.09.011> for a recent review [4]), and is a mechanism to guide epigenetic modifiers like p300 to particular genomic regions.

Reviewer figure 10: Identification and validation of protein interacting with *Sarrah*.

- a. *Sarrah* pulldown was performed by adding 200 pmol of biotinylated scrambled oligo or two biotinylated *Sarrah* antisense oligos to HL-1 cell lysate, coupling to streptavidin beads and elution. Pulldown efficiency was determined by qRT-PCR of eluted samples (displayed as % input; n = 6; SEM; * t-test p < 0.05).
- b. Volcano plot showing all proteins identified by mass spectrometry analysis that are enriched in *Sarrah* pulldown as compared to pulldown with a scrambled oligo. CRIP2, the first hit, is highlighted.
- c. RNA-immunoprecipitations with antibodies against CRIP2 (rabbit antibody), p300 (mouse antibody) and histone acetylation H3K27ac (rabbit antibody) were performed in primary human cardiomyocytes. *Sarrah* levels were measured by qRT-PCR (n = 4-8; SEM; * t-test p < 0.05; ** t-test p < 0.01; IgG: immunoglobulin G).

11. Figure 4l. Overexpression of mouse *Sarrah* lacking the triple helix domain increases apoptosis in mouse cardiomyocytes. This is surprising. Overexpression of WT *Sarrah* should decrease apoptosis (this control should be included in the experiment), and possibly delta *Sarrah* should not produce any effects on cardiomyocyte apoptosis but there are no obvious reasons for the deleted transcript to increase apoptosis.

We apologize for the confusion. The transcript containing the deletion, which we named *Sarrah* Δ TH, is not overexpressed, but transcribed from the endogenous locus that we mutated using a CRISPR/Cas9-based approach. The deletion of the triple helix forming region of *Sarrah* is now

described more precisely in the *Materials and Methods* section as well as in the figure legend. An increase in apoptosis with endogenously expressed *Sarrah* lacking its functional region corresponds to the effects observed after *Sarrah* knockdown (fig. 3h, supplementary fig. 5e and reviewer fig. 11).

Reviewer figure 11: Deletion of the endogenous *Sarrah* triple helix domain increases apoptosis.

- Scheme showing the CRISPR/Cas9-mediated approach to delete the endogenous *Sarrah* triple helix domain (TH) from mouse cardiomyocytes (HL-1 cell line).
- Apoptosis in mutated HL-1 cells from (a) was quantified as caspase-3/7 activity (n = 3; SEM; ** t-test p < 0.01).

Additionally, we included a caspase assay with *Sarrah* overexpressing human cardiomyocytes as suggested by the reviewer. The assay shows a profound decline of caspase-3/7 activity upon *Sarrah* overexpression and supports our *in vivo* findings that *Sarrah* overexpression has an anti-apoptotic effect on cardiomyocytes (fig. 1f and reviewer figure 12).

Reviewer figure 12: *Sarrah* overexpression in cardiomyocytes decreases apoptosis.

- Primary human cardiomyocytes were transduced with lentiviral vectors. *Sarrah* overexpression was assessed by qRT-PCR (n = 4; SEM; * t-test p < 0.05).
- Caspase-3/7 activity was measured in *Sarrah* overexpressing primary human cardiomyocytes (n = 4; SEM; * t-test p < 0.05).

12. The target loci of *Sarrah* action should be investigated using a ChIRP approach.

We thank the reviewer for the suggestion. To address this remark, we first validated *Sarrah* association with genomic DNA by RIP using the S9.6 antibody [5] that recognizes RNA:DNA hybrids. Our results show that *Sarrah* is indeed enriched at these structures as compared to the U4 snRNA negative control (fig. 3i and reviewer fig. 13a). In a second experiment, we performed ChIP using the same antibody and measured *Sarrah* target gene promoters in genomic DNA fragments associated with triple helices by qRT-PCR. All five target genes were, unlike the GAPDH promoter

negative control, enriched over IgG control. Four of the genes were significantly enriched over IgG control after RNase H digestion, which degrades RNA-DNA-structures such as R loops without affecting triple helices [6] (fig. 3j and reviewer fig. 13b). We therefore conclude that the majority of predicted *Sarrah* target genes forms triple helices in their promoters.

Reviewer figure 13: *Sarrah* target genes form triple helices.

- c. RNA-immunoprecipitation with the S9.6 anti-DNA-RNA-hybrid antibody was performed in crosslinked primary human cardiomyocytes. Levels of U4 snRNA as a negative control and *Sarrah* were measured by qRT-PCR (n = 3; SEM; * t-test p < 0.05; IgG: immunoglobulin G).
- d. RNA-immunoprecipitation with the S9.6 anti-DNA-RNA-hybrid antibody was performed in crosslinked primary human cardiomyocytes. Sonicated DNA fragments were used for qRT-PCR to quantify triplex formation in gene promoters.

Minor

1. Often, the legends, and even the methods are not clear. For instance, given the variety of models used in the paper, the authors should carefully indicate in the legend which cell types and which species are studied in the various experimental approaches described in the paper. In particular, the authors should clearly indicate when a cell line is used and when they rely on primary cells. Several key methods are not described, for instance the Triplex Domain Finder analysis.

We thank the reviewer for this feedback and have rewritten the figure legends and methods in order to make our work easier understandable for the reader.

2. The protocol using total Histone 3 immunoprecipitation to detect potential association of *Sarrah* with chromatin (Fig. 4C) should include a step in which precipitated material is treated with DNase to avoid possible amplification of *Sarrah*-encoding DNA.

We fully agree on this valid remark and are aware of the necessity to exclude amplification of DNA. A DNase digestion step was performed during all RNA extractions for this manuscript, which we now state in the *Materials and Methods* section. To additionally ensure that no DNA is precipitated and amplified during qRT-PCR, we included a -RT control of the input (the sample expected to contain the highest DNA amount) in our cDNA synthesis reaction to which no reverse transcriptase was added. Ct values of -RT control compared to Ct values of samples indicate that no DNA has been amplified and that detected *Sarrah* signals result from reversely transcribed RNA.

- [1] D. Ramanujam, Y. Sassi, B. Lagerbauer, and S. Engelhardt, "Viral Vector-Based Targeting of miR-21 in Cardiac Nonmyocyte Cells Reduces Pathologic Remodeling of the Heart," *Mol. Ther.*, vol. 24, no. 11, pp. 1939–1948, 2016.
- [2] S. Diederichs, "The four dimensions of noncoding RNA conservation," *Trends Genet.*, vol. 30, no. 4, pp. 121–123, 2014.
- [3] D. F. Chang, N. S. Belaguli, J. Chang, and R. J. Schwartz, "LIM-only protein , CRP2 , switched on smooth muscle gene activity in adult cardiac myocytes," *Proc Natl Acad Sci U S A*, 2007.
- [4] Y. Li, J. Syed, and H. Sugiyama, "RNA-DNA Triplex Formation by Long Noncoding RNAs," *Cell Chem. Biol.*, vol. 23, no. 11, pp. 1325–1333, 2016.
- [5] S. J. Boguslawski, D. E. Smith, M. A. Michalak, K. E. Mickelson, C. O. Yehle, W. L. Patterson, and R. J. Carrico, "Characterization of monoclonal antibody to DNA • RNA and its application to immunodetection of hybrids," vol. 89, pp. 123–130, 1986.
- [6] A. Postepska-Igielska, A. Giwojna, L. Gasri-plotnitsky, N. Schmitt, A. Dold, D. Ginsberg, and I. Grummt, "LncRNA Khps1 Regulates Expression of the Proto- oncogene SPHK1 via Triplex-Mediated Changes in Chromatin Structure Article LncRNA Khps1 Regulates Expression of the Proto-oncogene SPHK1 via Triplex-Mediated Changes in Chromatin Structure," *Mol. Cell*, vol. 60, no. 4, pp. 626–636, 2015.

Reviewers' comments:

Reviewer #1 (Remarks to the Author):

This Reviewer remains unsatisfied with the shortcut the authors took to perform AAV gene delivery of Sarrah in young rather than prematurely aged mice. For the authors to claim that Sarrah is related to the ageing heart, a prematurely aged model (pending on approval of ethics protocols and requesting the Editor additional time to complete the manuscript revision) or naturally aged mice (purchase at regular commercial vendor) are the model of choice.

Similarly, the omission of a loss-of-function study is possibly an even bigger weakness. This Reviewer is sympathetic towards Gapmers not functioning, but alternative approaches could have been considered and tested (AAV delivery of a shRNA, CRISPR deletion of the transcriptional start site, Aptamers...). As the manuscript stands now, the main functional conclusions of the manuscript are merely derived from in vivo gain of function experiments (a supraphysiological condition that normally doesn't occur for Sarrah) performed in the wrong mouse model.

A third outstanding item deals with the experimental validation of Sarrah downstream targets: the authors rebuttal with undisclosed data that 5 selected targets from the 135 identified did not reveal a phenotype in terms of apoptosis and stop shortly there. How were those 5 targets selected? Where the selection criteria ill-designed, could testing of other targets unveil the mechanism, in fact, what is the mechanism of action of Sarrah altogether?

Reviewer #2 (Remarks to the Author):

This new version of the manuscript by Trembinski et al. has been substantially improved. There are nevertheless a couple of issues that need to be addressed.

The authors make a strong point about the role of Sarrah in the regulation of contractility but this is not substantiated by data. Contractility is measured in isolated cardiomyocytes, in EHT and in vivo after Sarrah knockdown, and found to be significantly affected. However, this could be secondary of induced apoptosis in the myocyte population. This problem was raised previously and, in their

rebuttal, the authors do not provide a convincing argument in favor of a direct effect of Sarrah on mechanisms controlling contractility. In vivo and in EHT, decreased contractility might simply reflect a reduction in the number of contractile units (cardiomyocytes). Even in isolated cardiomyocytes, one cannot exclude that dying cells upon Sarrah knockdown are less functional (contractile) than healthy cardiomyocytes. The authors should investigate further this potential mechanism. One way could be to identify Sarrah-regulated target genes relevant to contractility (PDE3A is one example but not enough to explain the observed effects), and test modulation of the associated pathways upon Sarrah knockdown. Alternatively, the authors should adjust their message.

Minor points

1. Sarrah is expressed in both cardiomyocytes and endothelial cells, and then downregulated in the heart after infarction. However, the authors did not measure cell-specific expression after infarction. Expression should therefore be measured in cardiomyocytes and in endothelial cells before and after infarction.
2. Screenshots of the UCSC browser showing the mouse and human Sarrah locus should be included in the paper.
3. Suppl. Fig. 2B: The color code for the bar graph is not included
4. Capital letters should be used for human SARRAH

Reviewer #1 (Remarks to the Author):

This Reviewer remains unsatisfied with the shortcut the authors took to perform AAV gene delivery of *Sarrah* in young rather than prematurely aged mice. For the authors to claim that *Sarrah* is related to the ageing heart, a prematurely aged model (pending on approval of ethics protocols and requesting the Editor additional time to complete the manuscript revision) or naturally aged mice (purchase at regular commercial vendor) are the model of choice.

Response: At the reviewer's request we have performed AAV9-mediated overexpression of *Sarrah* or control (GFP) in naturally aged mice (18 months old) and assessed cell death by TUNEL staining (figure 5A). *Sarrah* overexpression reduced cell death by ~40%. These important experiments show that *Sarrah* augmentation in aged mice reduces ongoing age-induced apoptosis of cardiomyocytes.

Similarly, the omission of a loss-of-function study is possibly an even bigger weakness. This Reviewer is sympathetic towards Gapmers not functioning, but alternative approaches could have been considered and tested (AAV delivery of a shRNA, CRISPR deletion of the transcriptional start site, Aptamers...). As the manuscript stands now, the main functional conclusions of the manuscript are merely derived from in vivo gain of function experiments (a supraphysiological condition that normally doesn't occur for *Sarrah*) performed in the wrong mouse model.

Response: After consultation with the editor, we decided not to further pursue a loss-of-function model, as the costs/risks of developing such a model do not outweigh the potential gain in insights into the mechanism by which *Sarrah* regulates cardiomyocyte function, for the following reasons:

- 1. We have already tried several expensive gapmers, which did not work (as outlined in our previous response to reviewers). We also tested AAV-delivered shRNA, but using this delivery route does not result in active siRNAs that reduce nuclear *Sarrah* levels. So this approach also does not work.**
- 2. Generating a *Sarrah* knock-out mouse is technically very challenging, since we need to make sure we delete *Sarrah* without affecting the overlapping gene *Oxct1*. This would be a very high risk investment.**
- 3. If we were to construct a *Sarrah* knock-out mouse this would cost at least another year and >40.000€.**
- 4. What would be the added insight of in vivo loss-of-function model? We already provide ample experiments with *Sarrah* loss-of-function in mouse, human and rat cells. Furthermore, we show experiments with loss of the TH domain in *Sarrah* and last but not least: *Sarrah* loss-of-function in human engineered heart tissue, arguably even more closely mimicking patient heart muscle than a rodent model.**
- 5. In light of therapeutic relevance, a gain-of-function model (which would be the preferred therapeutic strategy in patients) gives much more valuable information than a loss-of-function model.**

Therefore, the very risky investment to establish a *Sarrah* loss-of-function mouse does not outweigh the benefit of showing that deletion of *Sarrah* induces cardiomyocyte apoptosis (something we have shown in at least 5 different models already). The editor agreed with us and we have discussed the limitations of not including an in vivo loss-of-function model in the discussion section of the manuscript.

A third outstanding item deals with the experimental validation of *Sarrah* downstream targets: the authors rebuttal with undisclosed data that 5 selected targets from the 135 identified did not reveal a phenotype in terms of apoptosis and stop shortly there. How were those 5 targets selected? Where the

selection criteria ill-designed, could testing of other targets unveil the mechanism, in fact, what is the mechanism of action of Sarrah altogether?

Response: We appreciate this request for more detailed mechanistic studies. We have addressed this question with multiple complimentary experiments. First, we took an unbiased approach to assess how SARRAH regulates apoptosis, using a commercially available proteome profiler array focused on apoptosis (figure 4A). This assay revealed an induction of pro-apoptotic proteins and a reduction of anti-apoptotic proteins after SARRAH depletion in human cardiomyocytes. Among the most profoundly downregulated proteins were catalase, bcl-2, heme oxygenase 1, and bcl-x. All four proteins are transcriptionally regulated by the Nrf2/Keap1 antioxidant response pathway. Interestingly, Nrf2 (*Nfe2i2*) is a predicted direct target of SARRAH and we were able to confirm that Nrf2 is reduced after SARRAH depletion (figure 4B). Furthermore, reactive oxygen species (ROS) levels are increased after SARRAH depletion, pointing to a reduction in anti-oxidant response (figure 4C). Finally, we assessed whether restoration of Nrf2 signaling by lentiviral overexpression of Nrf2 rescues the pro-apoptotic effect of SARRAH depletion (figure 4D). Indeed, Nrf2 overexpression partially negates the induction of apoptosis after SARRAH depletion, indicating that Nrf2 is one of the main mediators of the cardioprotective effects of SARRAH. Nonetheless, other SARRAH target genes likely contribute to the full cardioprotective phenotype controlled by SARRAH. We have included these results and critically discuss them in the revised manuscript.

Reviewer #2 (Remarks to the Author):

This new version of the manuscript by Trembinski et al. has been substantially improved. There are nevertheless a couple of issues that need to be addressed.

The authors make a strong point about the role of Sarrah in the regulation of contractility but this is not substantiated by data. Contractility is measured in isolated cardiomyocytes, in EHT and in vivo after Sarrah knockdown, and found to be significantly affected. However, this could be secondary of induced apoptosis in the myocyte population. This problem was raised previously and, in their rebuttal, the authors do not provide a convincing argument in favor of a direct effect of Sarrah on mechanisms controlling contractility. In vivo and in EHT, decreased contractility might simply reflect a reduction in the number of contractile units (cardiomyocytes). Even in isolated cardiomyocytes, one cannot exclude that dying cells upon Sarrah knockdown are less functional (contractile) than healthy cardiomyocytes. The authors should investigate further this potential mechanism. One way could be to identify Sarrah-regulated target genes relevant to contractility (PDE3A is one example but not enough to explain the observed effects), and test modulation of the associated pathways upon Sarah knockdown. Alternatively, the authors should adjust their message.

Response: We agree with the reviewer that it cannot be excluded that the contractile function as regulated by Sarrah, is secondary to a role in cell survival. We have therefore adjusted the manuscript accordingly and adjusted the results and discussion sections.

Minor points

1. *Sarrah* is expressed in both cardiomyocytes and endothelial cells, and then downregulated in the heart after infarction. However, the authors did not measure cell-specific expression after infarction. Expression should therefore be measured in cardiomyocytes and in endothelial cells before and after infarction.

Response: To address this point, we used RNA sequencing data recently published by us (Rogg et al., Circulation 2018) and assessed expression of *Sarrah* in endothelial cells, fibroblasts and cardiomyocytes isolated from mouse hearts 3 days after acute myocardial infarction or sham operated animals (supplementary figure 8A). These data confirm that *Sarrah* is highest expressed in cardiomyocytes, but is also present in endothelial cells and in fibroblasts. Furthermore, the data show that *Sarrah* levels are similarly reduced by approximately 2-fold in all cell types.

2. Screenshots of the UCSC browser showing the mouse and human *Sarrah* locus should be included in the paper.

Response: We agree with the reviewer and have included screenshots (supplementary figure 2).

3. Suppl. Fig. 2B: the color code for the bar graph is not included

Response: We have included a legend that denotes the different bar shades.

4. Capital letters should be used for human *SARRAH*

Response: We have changed mentions of human *SARRAH* with capitals.

REVIEWERS' COMMENTS:

Reviewer #1 (Remarks to the Author):

no further questions

Reviewer #2 (Remarks to the Author):

No further comments. I'd like to thank the authors for the opportunity to review this interesting work.